# Exocyst-mediated membrane trafficking of the lissencephaly-associated ECM receptor dystroglycan is required for proper brain compartmentalization

**Andriy S Yatsenko[1†], Mariya M Kucherenko[2†‡§], Yuanbin Xie[2#], Henning Urlaub[3,4], Halyna R Shcherbata[1,2]\***

[1]Institute of Cell Biochemistry, Hannover Medical School, Hannover, Germany; [2]Max Planck Research Group of Gene Expression and Signaling, Max Planck Institute for Biophysical Chemistry, Göttingen, Germany; [3]Bioanalytical Mass Spectrometry Research Group, Max Planck Institute for Biophysical Chemistry, Göttingen, Germany; [4]University Medical Center Göttingen, Bioanalytics, Institute for Clinical Chemistry, Göttingen, Germany

**\*For correspondence:**
Shcherbata.Halyna@mh-hannover.de

[†]These authors contributed equally to this work

**Present address:** [‡]Departmentof Cardiothoracic and Vascular Surgery, German Heart Center Berlin, Berlin, Germany; [§]Institute of Physiology, Charité - University Medicine Berlin, Berlin, Germany; [#]Department of Biochemistry and Molecular Biology, School of Basic Medical Science, Gannan Medical University, Ganzhou, Republic of China

**Competing interests:** The authors declare that no competing interests exist.

**Abstract** To assemble a brain, differentiating neurons must make proper connections and establish specialized brain compartments. Abnormal levels of cell adhesion molecules disrupt these processes. Dystroglycan (Dg) is a major non-integrin cell adhesion receptor, deregulation of which is associated with dramatic neuroanatomical defects such as lissencephaly type II or cobblestone brain. The previously established *Drosophila* model for cobblestone lissencephaly was used to understand how Dg is regulated in the brain. During development, Dg has a spatiotemporally dynamic expression pattern, fine-tuning of which is crucial for accurate brain assembly. In addition, mass spectrometry analyses identified numerous components associated with Dg in neurons, including several proteins of the exocyst complex. Data show that exocyst-based membrane trafficking of Dg allows its distinct expression pattern, essential for proper brain morphogenesis. Further studies of the Dg neuronal interactome will allow identification of new factors involved in the development of dystroglycanopathies and advance disease diagnostics in humans.

## Introduction

The complexity of the brain is generated by multiple types of neurons that connect to each other in a specialized manner, which often depends on selective cell adhesion (*Jontes and Phillips, 2006*). Neurons expressing similar cell adhesion molecules cluster together to organize brain compartments that have distinct functions; even more, selective cell adhesion is also used for the establishment of synaptic connections, allowing neurons to communicate and transfer information (*Kucherenko and Shcherbata, 2013*). In addition, the extracellular matrix (ECM) offers a structural support for brain morphogenesis and provides an organized microenvironment where signaling pathways interact to control neural cell division, differentiation, and pathfinding (*Long and Huttner, 2019*). Significant alterations in brain structure and function are generated even by modest changes in the ECM composition and the quantities of cell adhesion molecules on the neuronal cell surfaces. Importantly, multiple human neurodevelopmental diseases, which include, among many others, schizophrenia, autism, lissencephaly, learning disability, and language disorders, are caused by deficiencies of major cell adhesion molecules (*Edeleva and Shcherbata, 2013*; *Long and Huttner, 2019*). Therefore,

during differential neurogenesis, the spatiotemporal expression of cell adhesion and ECM proteins must be precisely regulated for proper brain assembly and function.

Dystroglycan (Dg) is a major non-integrin cell adhesion factor best known as a key component of the dystrophin glycoprotein complex (DGC), dysfunction of which is associated with a variety of muscular dystrophies (*Brancaccio, 2019*; *Matsumura and Campbell, 1994*; *Moore and Winder, 2010*). Importantly, while loss of dystrophin causes muscular dystrophies, such as Duchenne and Becker, abnormal function of Dg leads to congenital muscular dystrophies or dystroglycanopathies, such as muscle–eye–brain disease, the Walker–Warburg syndrome, and Fukuyama congenital muscular dystrophy, which are rare congenital syndromes with brain defects. Patients with dystroglycanopathy experience shortened lifespan, cognitive impairment and learning disability, refractory epilepsy, and hypotonia (*Michele et al., 2002*; *Waite et al., 2012*), which are accompanied by structural brain malformations called cobblestone brain. This dramatic phenotype is characterized by irregular borders, dysplasia, hypoplasia, and demyelination due to overmigration of neurons and glial cells beyond the external basement membrane (*Bönnemann et al., 2014*; *Michele et al., 2002*; *Moore et al., 2002*; *Nickolls and Bönnemann, 2018*; *Schiff et al., 2017*; *Waite et al., 2012*). Although it is known that clinically diagnosed congenital muscular dystrophies are caused by abnormal Dg functioning, only 30% of the dystroglycanopathy cases have been molecularly or genetically diagnosed (*ENMC DGpathy Study Group et al., 2017*). In vertebrates, Dg is implicated in multiple biological processes (*Herrador et al., 2019*; *Leonoudakis et al., 2014*; *Leonoudakis et al., 2010*; *Lindenmaier et al., 2019*; *McClenahan et al., 2016*; *Rambukkana et al., 1998*; *Sirour et al., 2011*; *Snow and Henry, 2009*); therefore, it is reasonable to expect that it has various interacting partners and is highly regulated. However, the current understanding of the factors influencing Dg functionality, especially in the brain, is limited.

Previous data revealed the importance of the precision of Dg expression in the *Drosophila* nervous system, for example, lower levels of Dg slow down neuronal stem cell division and cause hyperthermic seizures and defective axonal pathfinding, while higher levels accelerate proliferation and perturb neuron differentiation (*Kucherenko et al., 2008*; *Marrone et al., 2011a*; *Marrone et al., 2011b*; *Shcherbata et al., 2007*; *Yatsenko et al., 2014b*). Dg deregulation in the brain affects the distribution of major cell adhesion proteins, altering the composition of the ECM. This in turn causes the formation of structures that outgrow the normal contour of the ECM-defined brain space, resulting in abnormal brain tissue assembly (*Yatsenko et al., 2014b*). This phenotype is similar to the brain cortex abnormalities associated with dystroglycanopathies in humans, demonstrating that *Drosophila* can serve as a good genetic model for these disorders.

Like mammalian brains, the *Drosophila* brain is compartmentalized; different compartments have specific functions and are formed by families of neurons of the central nervous system called lineages, which innervate only a certain set of neuropil compartments (*Hartenstein et al., 2020*; *Insect Brain Name Working Group et al., 2014*). The most prominent compartments are antennal lobes (AL), the mushroom body (MB), and the central complex (CX), which consists of several neuropils, among which are the fan-shaped body (FB), ellipsoid body (EB), and protocerebral bridge (PB) (*Figure 1A–C*).

Considering the evolutionary conservation of functional brain compartmentalization and the similarity of the observed dystroglycanopathy brain pathologies in humans and flies, the advantageous *Drosophila* cobblestone lissencephaly model (*Yatsenko et al., 2014b*) was used to get a deeper insight about the factors that contribute to Dg function in the nervous system. Firstly, neuroanatomical studies of the pre-adult and adult brains were performed to analyze the Dg expression pattern. In the developing brain, Dg expression is spatiotemporally dynamic. It is more abundant during the pre-adult stages of brain development. In particular, Dg is present in the axonal projections of differentiating neurons assembling various brain compartments, suggesting a function in neuropil formation. Secondly, brain anatomy of mutants that have abnormal Dg expression was analyzed. It revealed that proper levels of Dg are required for proper brain compartmentalization and fasciculation of various major neuropils. Thirdly, using mass spectrometry, the neuronal Dg interactome was determined. In silico analyses of identified factors demonstrated that the majority of them have human disease-linked homologs and have never been shown to interact with Dg. By clustering Dg interactors into functional groups, multiple proteins of the exocyst complex were found. Finally, the data show that in neurons the exocyst regulates Dg membrane trafficking. A genetic interaction approach demonstrated that components of the exocyst complex functionally interact with Dg in

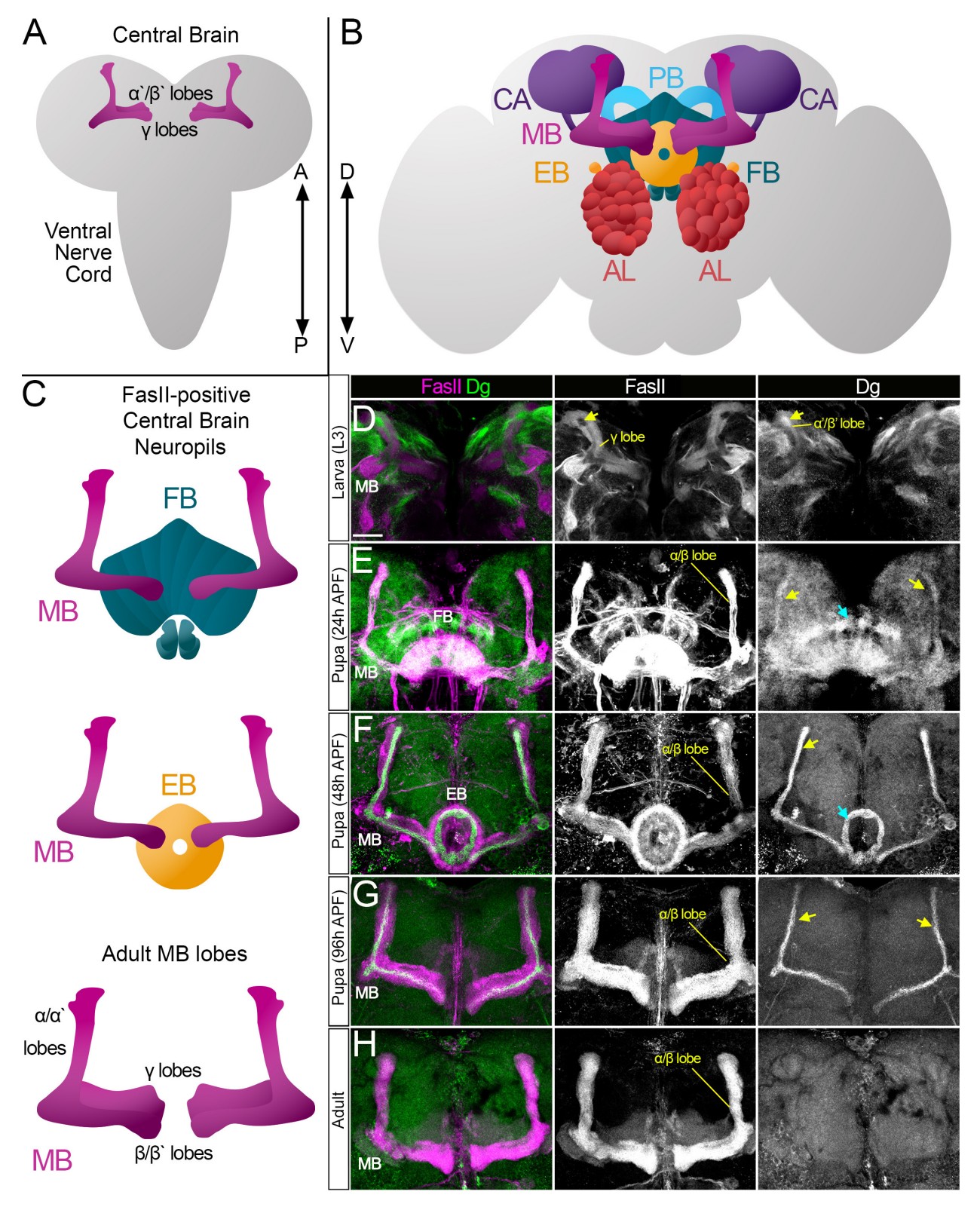

**Figure 1.** Survey of spatiotemporal dystroglycan (Dg) expression in the developing *Drosophila* brain. (**A–C**) Schematic representation of larval (**A**) and adult (**B**) *Drosophila* brains. Antennal lobes (AL, tomato), ellipsoid body (EB, orange), fan-shaped body (FB, teal), mushroom body (MB, fuchsia), calyx (CA, violet), and protocerebral bridge (PB, cyan) are shown. Major neuropils of the central brain are also shown separately (**C**). Anterior ←→ posterior (A←→P). Dorsal ←→ ventral (D←→V). (**D–H**) Anterior view of the midbrain region of the larval brain (**D**) and frontal views of the pupal (**E–G**) and adult

*Figure 1 continued on next page*

*Figure 1 continued*

(H) brains stained with antibodies against Dg (green) and the homophilic cell adhesion molecule Fasciclin II (FasII, magenta). Expression patterns for FasII and Dg are also shown in separate channels. Scale bar 20 μm. (D) In the larval brain, FasII is expressed in the γ lobes of the MB neuropil, which are formed in early larval stages. At L3 stage, α′/β′ lobe formation takes place. These lobes can be seen by Dg expression (arrow) and the absence of FasII expression. (E) After the pupa is formed, MB neuroblasts give rise to α/β lobe neurons that are positive for FasII. Dg expression is observed in newly generated neurons of α/β lobes (yellow arrows). In addition, a distinct Dg pattern is seen in neurons forming the FB (blue arrow) neuropil. (F) At mid-pupal stage, Dg expression remains in inner MB α/β lobe neurons (freshly generated differentiating axons, yellow arrow) and disappears from outer α/β neurons, which were born at earlier pupal stages (FasII marker demonstrates their belonging to α/β MB lobe). In addition, the Dg pattern diminishes from the FB but appears in the developing EB (blue arrow). (G) In final pupal stages, when most neuropils, except for MB α/β lobe neurons, are established, Dg protein is enriched in a small subset of inner α/β lobe axons (yellow arrows) and significantly reduced in other neuropils. Note the diminished Dg staining in fully formed FB and EB neuropils. (H) In the adult brain, Dg expression is visibly reduced.

brains. Moreover, there are temporal requirements in exocyst–Dg regulation for the establishment of brain compartments. Since exocyst–Dg regulation depends on the developmental stage, these findings propose that neuronal cells have different requirements for exocyst-regulated Dg trafficking at various stages of differentiation. Further analysis of identified neuronal Dg interactors in a *Drosophila* model should help to decipher neural-specific molecular functions of this key ECM receptor and provide important insights into the molecular mechanisms leading to the development of congenital muscular dystrophies in humans.

## Results

### In the developing brain, Dg is dynamically expressed in differentiating neuropils

To get an insight into the temporal dynamics of Dg expression in the pre-adult brain, the Dg protein localization was analyzed in the same midbrain region at larval (L3), pupal (24 and 48 hr after puparium formation [APF]), and pharate (96 hr APF) developmental stages. In the larval central brain (CB), Dg is expressed in patches of axonal projections (*Figure 1D*). Some fascicles can be visualized by antibodies raised against the homophilic cell adhesion molecule Fasciclin II (FasII). The largest FasII-expressing CB structure is the MB, which contains three sequentially produced neuronal subtypes whose axons cluster differentially to form MB lobes. First, the γ lobe is produced (FasII-positive), its neurons born during embryonic and early larval stages; next, α′/β′ lobe neurons are generated during mid-late larval stages (FasII-negative); and finally, the α/β lobe (FasII-positive) is formed during pupal stages (*Ito and Hotta, 1992*; *Lee et al., 1999*). At the L3 larval stage, Dg is seen in FasII-negative MB neurons, corresponding to α′/β′ lobe neurons (*Figure 1D*).

At the early pupal stage, formation of the FB neuropil and production of MB α/β lobe neurons take place (*Figure 1E*). When compared to larval stages, Dg expression at the early pupal stage has a more uniform pattern (Dg-negative patches disappear), and an increase in Dg in newly developing neuropils is evident (*Figure 1E*). Note that at this developmental stage the α/β lobe begins to be formed, which is marked by the appearance of FasII-positive axonal projections, and Dg expression coincides with most FasII-expressing neurons. In the next 24 hr, multiple neurons complete their differentiation, and at 48 hr APF, the Dg expression pattern changes again: it gets weaker in most of the midbrain and becomes very distinctive in certain neuropils (*Figure 1F*). Dg protein diminishes from the FB and appears in the EB neuropil, which is formed at mid-pupal stage.

Moreover, during later pupal stages, Dg's expression pattern in the α/β MB lobe no longer fully overlaps with FasII staining but is rather in a thin pattern positioned at the center of the FasII-positive fascicle (*Figure 1F, G*). During MB lobe formation, the most recently born neurons extend their axons along the cross-sectional center of the lobe such that they are surrounded by older, more differentiated neurons (*Kunz et al., 2012*; *Kurusu et al., 2002*; *Sinakevitch et al., 2010*). The presence of Dg in the center of the α/β lobe (*Figure 1F*) is consistent with its expression in differentiating neurons. At the pharate pre-adult stage (96 hr APF), when all CX neuropils, except for MB α/β neurons, have completed their differentiation (*Andrade et al., 2019*; *Lee et al., 1999*), Dg protein fades from the EB neuropil and persists exclusively in subsets of MB axons (*Figure 1G*).

In summary, this analysis shows that the Dg expression pattern has a spatially and temporally dynamic character. It is strongly expressed in axons of the recently born, differentiating neurons

(*Figure 1D–G*) and diminishes after the differentiation/maturation process is completed (*Figure 1H*). Thus, Dg's targeted expression in developing neuropils suggests that it could be involved in neuropil formation and brain compartmentalization. Therefore, the next experiments addressed whether and how these processes are affected upon Dg misexpression.

## Dg dysregulation affects architecture of neuropils

First, neuroanatomical studies of adult brains dissected from loss- and gain-of-function *Dg* mutants were performed. In particular, trans-allelic $Dg^{O55}/Dg^{O86}$ animals and mutants that had Dg overexpressed in the neurons, *insc>Dg* (for Dg protein expression, see *Figure 2—figure supplement 1*), were examined. Analyses of histological sections showed gross abnormalities in the brain organization, which included the lumpy brain surface and atypical compartmentalization of various brain neuropils (*Figure 2A–C*, arrows).

Importantly, the histochemical analyses of the brains dissected from the rare survivors with Dg deficiency or Dg neuron-specific upregulation demonstrate that in comparison to controls the appearance of their major midbrain neuropils is noticeably perturbed (*Figure 2D–F*). The most prominent structures that can be visualized in the frontal view of adult brains are the ALs and the MB. In particular, the olfactory memory centers (ALs) display abnormal shapes (*Figure 2G–I*, green arrows). Similarly, MB and PB neuropils show major abnormalities. The PB is positioned posteriorly from the protocerebral neuropil between the calyces of the MB. Normally, the PB is an elongated structure reminiscent of a bicycle handlebar with slightly ventrally bent ends. In *Dg* mutants, the PB handles are compressed toward the midline (*Figure 2H, I*, white arrows). Moreover, the MB of *Dg* mutants also appears highly disorganized; the form of MB calyces is altered (*Figure 2H, I*, magenta arrows), and the size and shape of the lobes as well as their neuronal projections look abnormal. For example, β- and β′ lobes often cross the midline (*Figure 2H*, yellow arrowheads) and α- and α′ lobes appear to be underdeveloped (*Figure 2H, I*, yellow arrows). These data suggest that Dg is involved in the establishment of brain compartments.

## Proper levels of Dg are important for MB fasciculation

Next, to measure the impact of Dg expression levels on neuropil formation, the architecture of the last-born α/β MB lobes was analyzed. These lobes could be marked by FasII and easily scored for distinct phenotypes (*Figure 3A*). In *Dg*-deficient brains, 20% of α/β lobes analyzed are misguided and more than 30% are underdeveloped (*Figure 3B, C, I*; for quantifications, see *Supplementary file 1*). These phenotypes are even more prominent in *Dg* gain-of-function mutants: less than 10% of α/β lobes appear normal, whereas 30% are dramatically underdeveloped and 60% are completely misguided with their neuronal projections emerging in atypical parts of the brain (*Figure 3D–F, I*, *Supplementary file 1*). These data suggest that the proper levels of Dg are critical for brain neuropil formation and axonal pathfinding.

To address if these phenotypes are due to an intrinsic requirement for Dg in MB neurons, Dg was up- and downregulated (*UAS-Dg* and *UAS-Dg$^{RNAi}$*) specifically in MB neurons using *c309-Gal4* (pan-MB + eye and antennal disc expression) and *201Y-Gal4* (γ and α/β) drivers (*Aso et al., 2009*). Importantly, Dg up- or downregulation in MB neurons results in the appearance of defects like those observed in *Dg* mutant α/β lobes: α/β lobes are significantly underdeveloped and often misguided (*Figure 3A, B*, *Supplementary file 1*). Notably, the phenotypes caused by Dg deregulation in MB neurons are less dramatic than the axon misguidance phenotypes observed upon Dg misexpression in the entire brain (compare *Figure 3D, F and G, H*), suggesting that for the proper assembly of the brain, Dg expression must be controlled in multiple neurons. Together, these data show that the precise expression of the ECM receptor Dg specifically in the developing neurons is critical for their proper axonal pathfinding and establishment of brain neuropils. Since these phenotypes are similar to the anatomical brain defects of human dystroglycanopathy patients, it suggests that analysis of Dg signaling in a *Drosophila* model could give important insights into the molecular mechanisms leading to the development of congenital muscular dystrophies.

## Identification of Dg interactome in neurons

*Drosophila* offers a unique possibility to study the Dg interactome since tagged Dg protein can be expressed in a tissue-specific manner to isolate Dg complexes. Previously, using this technique, Dg

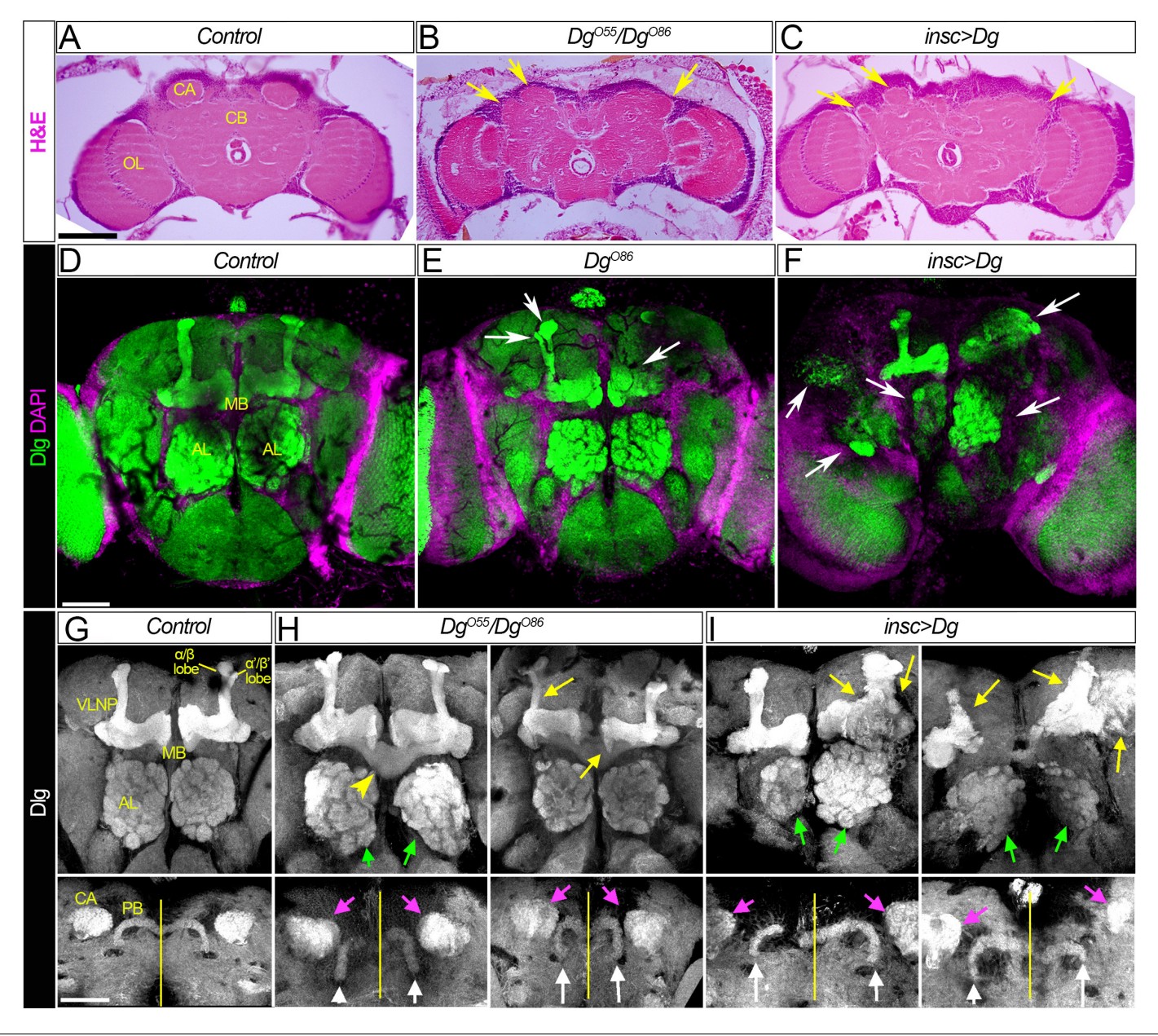

**Figure 2.** Dg is required for proper development of neuropils. (A–C) Hematoxylin and eosin (H&E) stained histological sections of adult brains of control, *Dg* loss- and gain-of-function mutants (A: *w1118/OregonR*; B: *DgO55/DgO86*; and C: *insc>Dg*). Note the appearance of the cobblestone brain phenotype in *Dg* mutants and abnormal formation of brain neuropils (yellow arrows). OL: optic lobe; CA: calyx; CB: central brain. (D–F) Frontal-anterior view of adult brains of control, *Dg* loss- and gain-of-function mutants (D: *w1118/OregonR*; E: *DgO86*; and F: *insc>Dg*). Anti-discs large (Dlg, green) antibody marks septate junctions and is used to label membranes of neuronal cell bodies, neuronal fibers, and synapses, while DAPI (magenta) marks nuclei. Note that midbrain neuropils are abnormal in mutants with deregulated dystroglycan (Dg) expression (white arrows). MB: mushroom body; AL: antennal lobe. (G–I) Frontal-anterior view of the central brain in control, *Dg* loss- and gain-of-function mutants (G: *w1118/OregonR*; H: *DgO55/DgO86*; and I: *insc>Dg*). Anti-Dlg – grayscale. Upper panels show the α/β (bright) and γ (dim) MB lobes marked with anti-Dlg marker. Note that upon Dg deregulation ALs and MB neuropils (green and yellow arrows, respectively) are disorganized. Lower panels show the frontal-posterior views of adult brains showing abnormal shape of MB calyces (CA, magenta arrows) and the protocerebral bridge (PB, white arrows) in *Dg* loss- and gain-of-function mutants in comparison to the control. Yellow vertical line shows the midline. Scale bar 50 μm.

The online version of this article includes the following figure supplement(s) for figure 2:

**Figure supplement 1.** Dystroglycan (Dg) expression patterns.

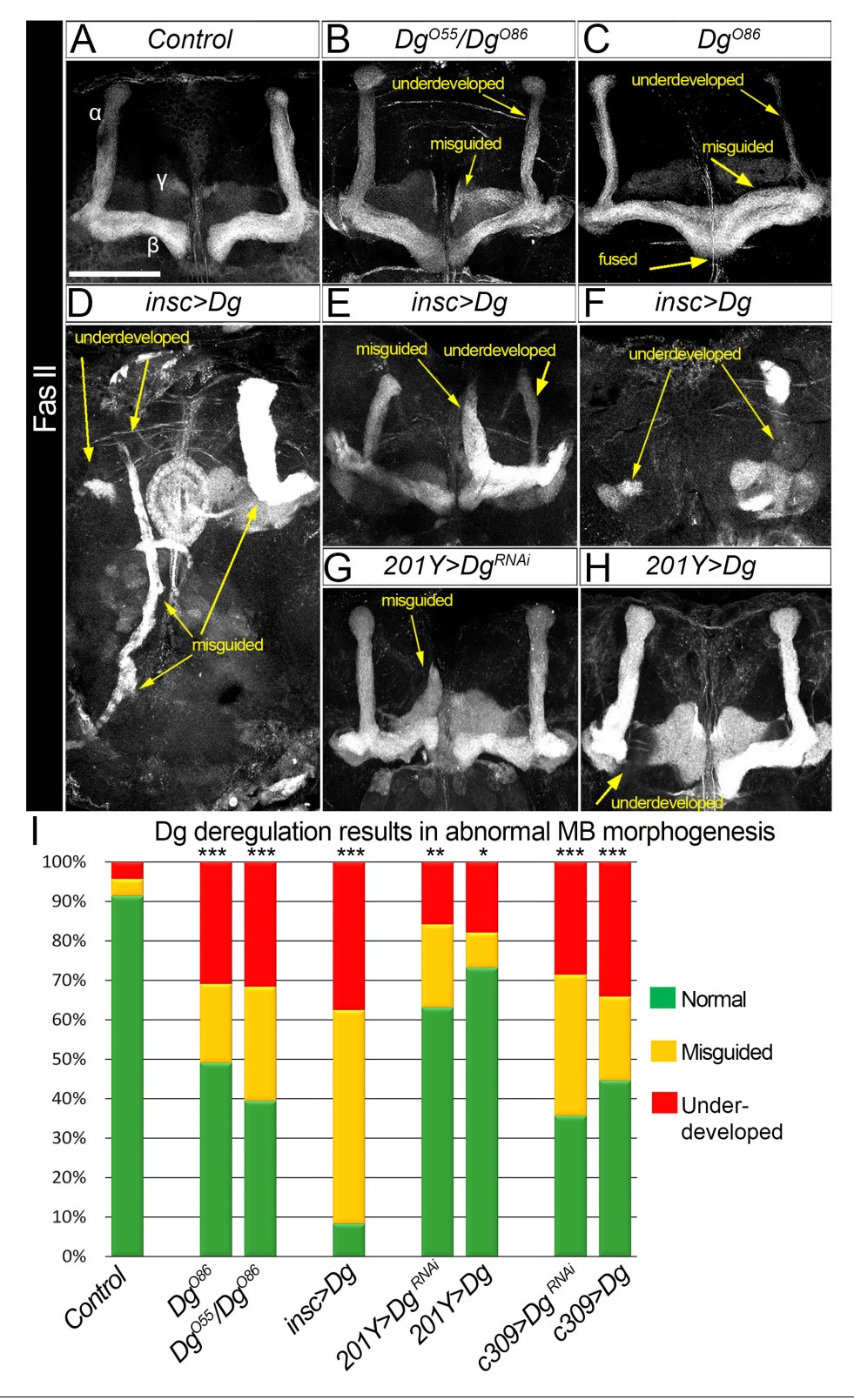

**Figure 3.** Dystroglycan (Dg) is required for proper mushroom body (MB) fasciculation. (**A–H**) Fasciclin II (FasII) staining of adult mutant brains reveals various morphological α/β lobe defects upon Dg deficiency (**B**: $Dg^{O55}/Dg^{O86}$; **C**: $Dg^{O86}$), pan-neuronal Dg upregulation (*insc>Dg*, **D–F**), and γ and α/β MB neuron-specific Dg down- or upregulation (*201Y>Dg^{RNAi}*, **G**, and *201Y>Dg*, **H**). In Dg mutants, axons of α lobe neurons stop migration prematurely or abnormally project into β lobe space, forming underdeveloped α lobes. Axons of β lobe neurons are improperly clustered and misguided,

*Figure 3 continued on next page*

*Figure 3 continued*

projecting into γ lobe space or overshooting the midline to form a fused β lobe. Note that misguided and underdeveloped α/β lobe phenotypes are more dramatic when Dg is overexpressed in all neuronal cells. (I) Quantification of the observed MB phenotypes (see also *Supplementary file 1*). For comparison of the observed phenotypes, $\chi^2$ test was used. ***$p \leq 0.001$; **$p \leq 0.01$; *$p \leq 0.05$; n.s.: not significantly different.

interactors were identified in muscles (*Yatsenko et al., 2020*). Here, the idea was to isolate Dg interactors in neurons. Because continuous overexpression of Dg during development with pan-neuronal drivers is lethal (*Yatsenko et al., 2014b*), a pilot screen to identify Dg interactions just in the adult nervous system was performed. To avoid the high lethality rate caused by Dg overexpression during development, Dg was overexpressed only in adult animals by using the switchable *Gal4/Gal80^{ts}* genetic system. In particular, GFP-tagged, full-length Dg was overexpressed in adult neuronal cells using the pan-neuronal driver *elav-Gal4.* Young adults were kept for 5 days at restrictive temperature (29°C) to ensure the sufficient expression of the tagged protein for mass spectrometry (for details, see Materials and methods). Dg complexes from adult *Drosophila* heads were immune-isolated in order to detect precipitated proteins by mass spectrometry analysis (*Figure 4A*). Results were verified in duplicates (*Figure 4B*), and all immunoprecipitated proteins enriched at least twofold in comparison to controls are reported as Dg interactors (*Figure 4B, C*).

To gain insights into the biological roles of Dg in the central nervous system (CNS) and the pathways with which it interacts in this system, intensive bioinformatic analyses of identified proteins were performed to address their cellular localization (*Figure 4C*), molecular function, and associated biological processes (*Figure 4D*), as well as human homologs and disease associations (*Supplementary file 2*). Importantly, the human disease-association enrichment analysis identified that human homologs of Dg-interacting proteins detected in this study have significantly enriched associations with nervous system diseases and mental disorders (*Supplementary file 3*). These analyses distinguished functional groups that include ECM components; proteins that mediate membrane transport; regulators of synaptic and other types of cytoplasmic vesicles; elements associated with ER, Golgi, or mitochondria; nuclear envelope factors; components that regulate protein degradation; nuclear and cytoplasmic RNA exosome complexes; and chromatin remodeling factors (*Figure 4C*, *Supplementary files 2* and *4*). To better characterize the identified factors, they were placed in a protein interaction network, grouped based on the reported molecular functions and clustered into protein complexes by utilizing the Markov clustering algorithm (MCL, *Figure 4D*, *Supplementary file 5*).

## Dg functional interaction network includes disease-associated elements

Currently, few interactors have been identified as Dg interactors in any organism, especially in the nervous system. Apart from several components of the DGC per se, only a small number of ECM proteins, such as agrin, pikachurin, perlecan, and laminins, have been shown to bind to Dg. Encouragingly, a *Drosophila* homolog of agrin, eyes shut (Eys), was also detected in the protein interaction network. It clustered together with the other two known Dg interactors, Kibra and Vimar (*Figure 4D*, red, see also *Supplementary files 2* and *5*; *Kucherenko et al., 2011*; *Marrone et al., 2011a*; *Yatsenko et al., 2020*). This group includes several proteins that play a role in eye morphogenesis in *Drosophila* and humans (*Jukam and Desplan, 2011*; *Mahato et al., 2018*; *Marrone et al., 2011a*; *Ray et al., 2020*). Moreover, these factors are not only associated with various ocular dystrophies but also involved in cancer development and regulation of stem cell maintenance and differentiation, cell growth, and metabolism (*Cehajic-Kapetanovic et al., 2019*; *Collison et al., 2019*; *Jamal et al., 2020*; *Lu et al., 2019*; *Priedigkeit et al., 2021*; *Wu et al., 2020*). For example, in *Drosophila,* prominin (Prom), a homolog of human CD133, maintains mitochondrial function, regulates body size and weight, and influences animal longevity by controlling insulin and TOR signaling (*Ryu et al., 2019*; *Wang et al., 2019*; *Zheng et al., 2019*). These functions resonate with the previously described functions for Dg in control of neuronal stem cell proliferation and differentiation, establishment of cellular polarity, maintenance of cellular homeostasis, and – at the organismal level – control of embryogenesis, stress response, adult animal metabolism, and longevity (*Kreipke et al., 2017*;

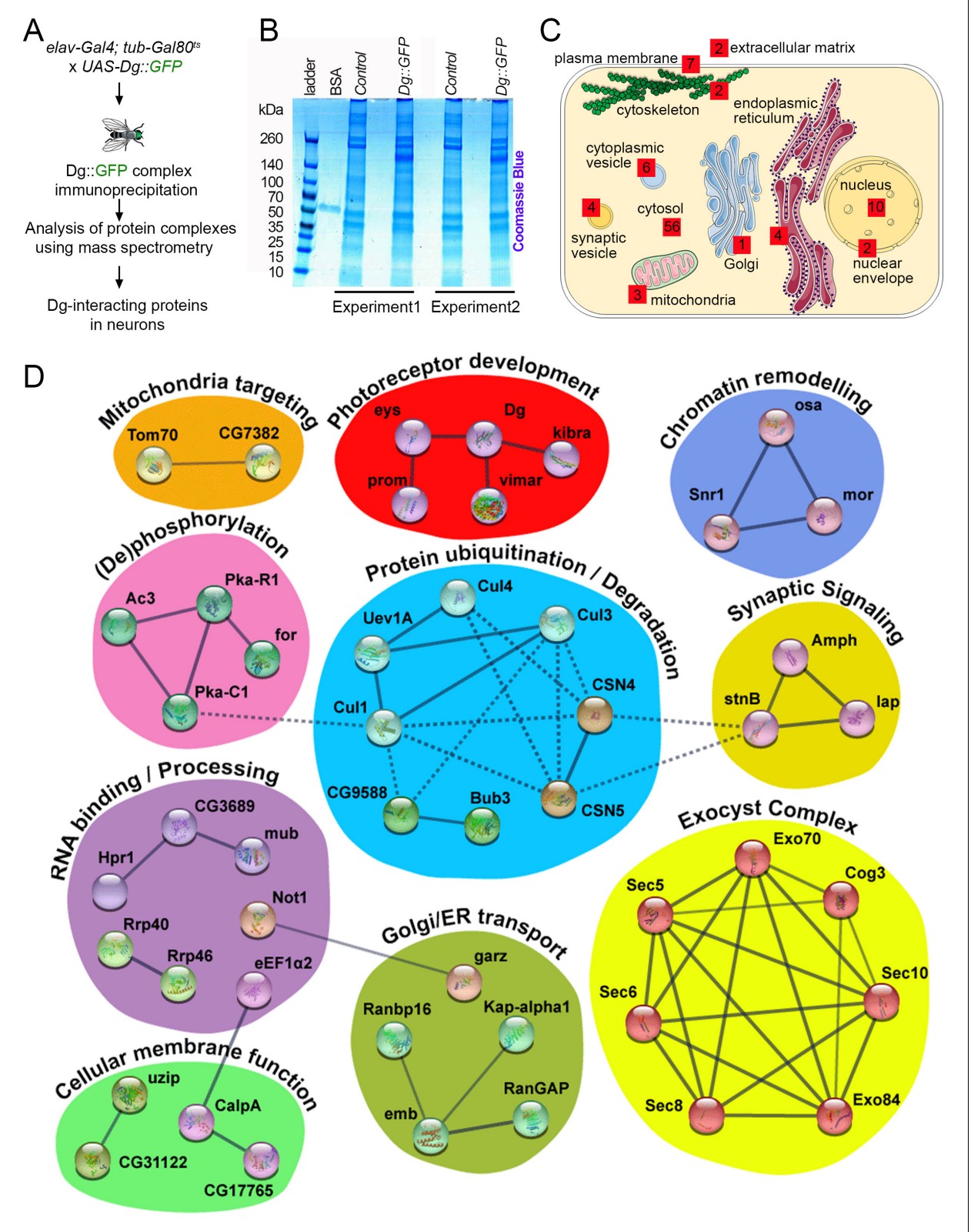

**Figure 4.** Neuronal dystroglycan (Dg)-associated components identified through proteomics approach. (**A**) Scheme represents experimental techniques carried out to identify neuronal proteins that interact with Dg. GFP-tagged full-length Dg was expressed specifically in neurons by driving expression of *UAS-Dg::GFP* with *elav-Gal4* using the *Gal4/Gal80^ts* system. Dg::GFP protein was immunoprecipitated with GFP-Trap beads containing anti-GFP antibodies. Proteins that form complexes with Dg in neuronal tissue were detected by mass spectrometry analysis. (**B**) Coomassie blue-stained gel

*Figure 4 continued on next page*

*Figure 4 continued*
confirms increased protein levels in samples immunoprecipitated from protein extracts from Dg-overexpressing adult animal heads. Experiments were performed in duplicate. (C) Cartoon represents neuronal cell with subcellular compartments, where red squares and numbers indicate identified Dg-associated proteins and their reported subcellular localization. See also *Supplementary files 2* and *4*. (D) Dg-associated components placed into a protein interaction network. Colored shapes outline functional groups. Nodes symbolize identified proteins, lines show previously reported associations, and line thickness represents confidence of association. Non-dashed lines show protein complexes identified by Markov clustering algorithm. See also *Supplementary file 5*. Source data file. Mass spec data for neuronal Dg interactome https://doi.org/10.5061/dryad.8sf7m0cmf.

*Kucherenko et al., 2010*; *Marrone et al., 2011b*; *Shcherbata et al., 2007*; *Yatsenko and Shcherbata, 2014*; *Yatsenko et al., 2014a*). Recently, Dg has been shown to interact with Kibra in both vertebrates and invertebrates (*Iyer et al., 2019*; *Morikawa et al., 2017*; *Vita et al., 2018*; *Yatsenko et al., 2020*). Kibra was also detected in this screen, suggesting that a Dg–Hippo signaling interaction might be also important in the nervous system. These data propose that in the nervous system the transmembrane protein Dg also acts as a scaffold that brings together different signaling components to the membrane. This idea is supported by previous studies in other tissues, for example, studies that show that Dg acts as a signaling hub in promoting nitric oxide syntase-histone deacethylase (NOS-HDAC) signaling, and Hippo and insulin signaling pathways in muscles (*Cacchiarelli et al., 2010*; *Eid Mutlak et al., 2020*; *Marrone et al., 2012*; *Marrone and Shcherbata, 2011*; *Vita et al., 2018*; *Watt et al., 2015*; *Yatsenko et al., 2020*; *Yatsenko et al., 2014b*).

In addition to signaling factors, mass spectrometry analysis allowed to identify components of the larger complexes that might play a role in Dg processing. For example, two important groups detected in this screen that may influence Dg intracellular interactions are the protein kinases and factors involved in protein degradation (*Figure 4D*, pink and blue). Previously, it was demonstrated that in *Drosophila* tyrosine phosphorylation of Dg at the C-terminal end prevents Dg binding to Dystrophin (*Yatsenko et al., 2007*; *Yatsenko et al., 2009*). Moreover, in vertebrates, Dg phosphorylation has been identified as a possible signal to promote the proteasomal degradation of the entire DGC (*Gazzerro et al., 2010*; *Lipscomb et al., 2016*; *Miller et al., 2012*; *Signorino et al., 2018*). Therefore, by preventing phosphorylation of Dg or inhibiting ubiquitination or proteasomal degradation, the DGC is stabilized (*Bachiller et al., 2020*; *ENMC DGpathy Study Group et al., 2017*). Studies in mice and zebrafish myoblasts and muscles have established that pharmacological treatment with proteasome or tyrosine kinase inhibitors can increase levels of non-phosphorylated Dg, which ameliorates the dystrophic phenotype (*Lipscomb et al., 2016*; *Vélez-Aguilera et al., 2018*). In the nervous system, however, the mechanisms of Dg phosphorylation, ubiquitination, and proteasomal degradation and the effects of these processes on neuronal differentiation and signaling have not been investigated. Therefore, it would be important to study in detail Dg interactions with the components of these two groups.

Interestingly, Dg interaction with agrins modulates the assembly of synapses (*Bassat et al., 2017*; *Fallon and Hall, 1994*; *Hilgenberg et al., 2009*; *Liu et al., 2020*), and apart from agrin, several proteins playing a role in synaptic signaling, such as Amphiphysin (Amph), Stoned B (stnB), and like-AP180 (lap), were identified (*Figure 4*, mustard). Similar to Dg, these regulators of synaptic vesicle transport and neurotransmitter secretion have been shown to be associated with neurodegeneration, myopathy, and cancer in humans (*Bao et al., 2005*; *Kumon et al., 2020*; *Narayan et al., 2020*; *Podufall et al., 2014*; *Taga et al., 2020*; *Xu et al., 2018*; *Zhang et al., 2020*). Glycosylated Dg is an essential organizer at various synapses, and depending on the brain region and cell type, Dg may function at presynaptic, postsynaptic, or glial sites of the synapse (*Früh et al., 2016*; *Nguyen et al., 2013*; *Noell et al., 2011*; *Orlandi et al., 2018*; *Satz et al., 2010*). For example, Dg ligands, agrin and neurexins, which are expressed at presynaptic terminals in the brain, may interact across the synapse with matriglycans on postsynaptic Dg (*Nickolls and Bönnemann, 2018*; *Yoshida-Moriguchi and Campbell, 2015*). However, the exact function of Dg as a part of a trans-synaptic protein complex, facilitating synapse formation and maintenance, remains unclear. Since it has already been shown that in vertebrates Dg mutants have impaired synaptic plasticity (*Kunz et al., 2012*; *Michele et al., 2002*; *Montanaro and Carbonetto, 2003*; *Moore et al., 2002*), further

analysis of Dg interaction with proteins involved in synaptic signaling would allow a better understanding of the role of Dg in synapse formation and signaling.

Also, there were several unexpected functional association groups, namely proteins of the Brahma complex involved in chromatin remodeling, mitochondria-targeting factors, membrane-associated receptors, and importins, functional interactions of which with Dg are less apparent and must be confirmed through follow-up experiments. In general, the biological value of the Dg protein interaction network identified here should provide prognosis of new Dg functions in the nervous system and aid in understanding complex phenotypes observed upon congenital dystrophies.

## Exocyst mediates delivery of Dg-carrying cytoplasmic vesicles to the plasma membrane

As the identified Dg neuronal interactome was largely novel, one of the most prominent functional groups was followed up to test whether the results of the mass spectrometric analysis have biological relevance. In particular, the focus was set on Dg association with exocyst complex proteins as this group was one of the most prominent clusters identified by the MCL algorithm. The exocyst is an octameric protein complex involved in tethering and spatial targeting of post-Golgi vesicles to the plasma membrane preceding SNARE-mediated fusion (*Ahmed et al., 2018*). It is evolutionarily conserved and involved in the regulation of multiple cell processes such as establishment of cell polarity, exocytosis, cell migration, and growth (*Langevin et al., 2005*). Of the eight subunits of the complex, the six proteins Sec5, Sec6, Sec8, Sec10, Exo70, and Exo84 were co-purified together with Dg (*Figure 4D*, *Supplementary file 4* and *5*). Considering that Dg is a transmembrane protein that is heavily glycosylated prior to its delivery to the membrane, the hypothesis can be put forward that the exocyst might be involved in mediating Dg's trafficking from the Golgi to the plasma membrane. Alternatively, Dg and exocyst complex proteins may interact at the membrane site. In this case, Dg could serve as a signal that guides secreting vesicle fusion to the specific location at the plasma membrane (*Figure 5A*).

First, to test whether loss of Dg in brain cells affects expression and/or localization of the exocyst complex, during early larval stages, $Dg^{O86}$ homozygous loss-of-function clones were induced, in which the expression pattern of a core component of the exocyst complex, Sec5 protein was analyzed (*Mott et al., 2003*). No obvious changes in either Sec5 protein expression levels or subcellular localization were observed in *Dg* mutant clones (visualized by the absence of GFP) in comparison to GFP-positive control areas in L3 larval brains (*Figure 5B*, *Figure 5—figure supplement 1*). This result suggests that exocyst-mediated vesicle trafficking is unlikely to be regulated by Dg protein.

Then, it was addressed whether the exocyst has an effect on Dg's subcellular localization. Loss-of-function mutations in exocyst complex subunits severely affect animal development and result in early larval lethality. Therefore, later-stage brains (L3 larval and pupal) containing clones mutant for exocyst complex components were analyzed. In *Sec5* and *Sec15* mutant clones in the brain, there is less membrane recruitment of Dg, but higher levels of Dg expression in intracellular puncta (*Figure 5C, D*).

Next, to investigate whether exocyst-dependent regulation of Dg expression indeed occurs in developing neurons, cells were marked with a neuron-specific anti-Elav antibody. Importantly, an abnormal Dg localization pattern was observed in *Sec5*-deficient neurons (*Figure 5D*). To assay whether the exocyst complex is required for the proper expression and membrane localization of Dg, *sec15* loss-of-function clones were induced in central brain neuroblasts (NBs) to analyze mutant NBs and their progeny. Dg appears to aggregate in the cytoplasm of neural progenitor cells as well as in their progeny, and Dg presence at the NB membrane is decreased in comparison to control (*Figure 5E*, arrows). These results suggest the involvement of Sec proteins in Dg regulation and propose that in the developing brain the exocyst complex mediates Dg protein trafficking to the plasma membrane of neuronal stem cells and differentiating neurons.

To further confirm that loss of exocyst function indeed causes a decrease in Dg at the cell membrane, Dg localization in the larger cells of the salivary glands was analyzed. In these cells, Dg and Sec5 proteins show clear co-localization at the membrane (*Figure 5—figure supplement 2A*). Clones deficient in the exocyst components Sec6 or Sec15 show a strong reduction in membrane-associated Dg (*Figure 5—figure supplement 2B–F*), implying that loss of exocyst function could

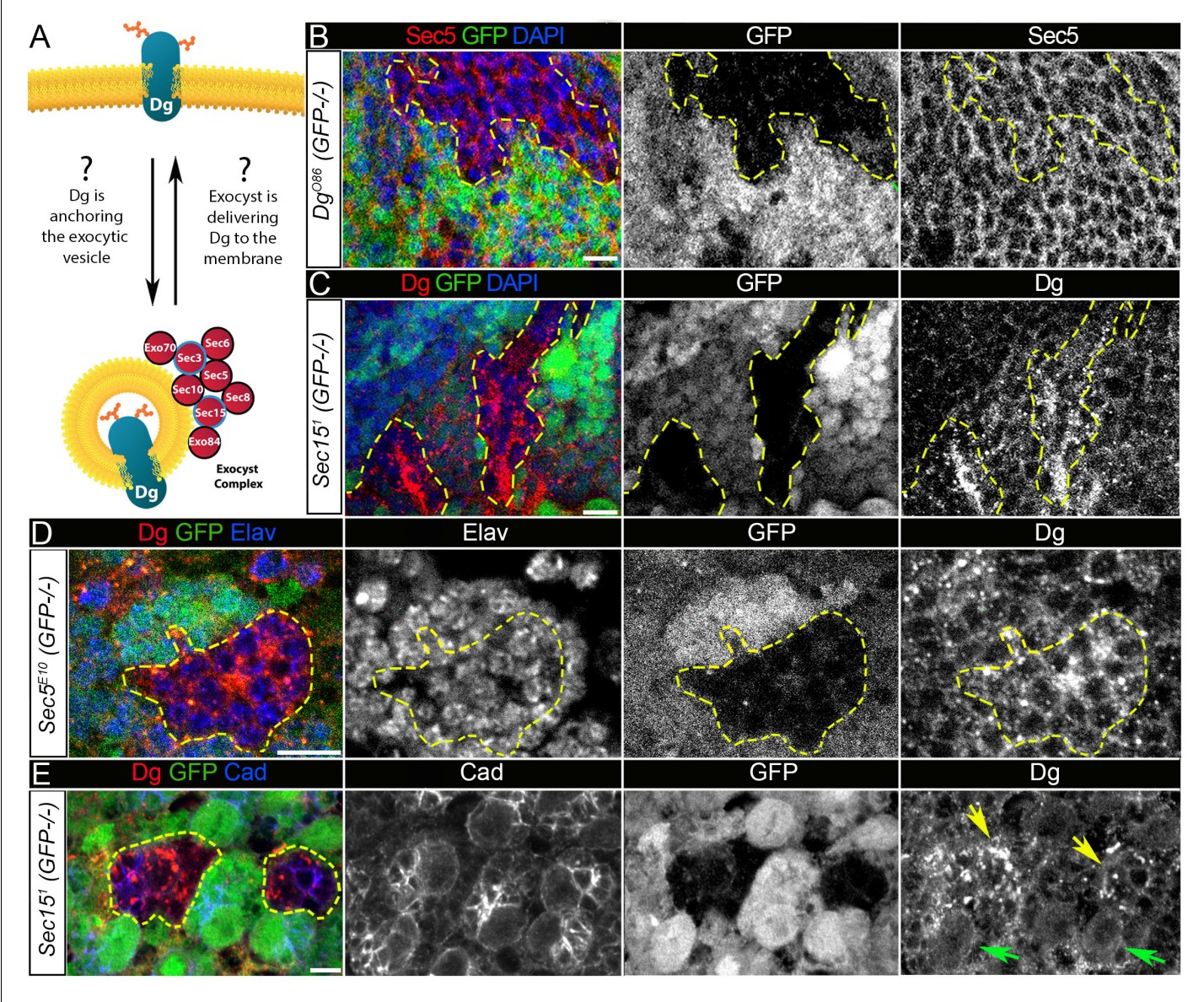

**Figure 5.** Exocyst mediates dystroglycan (Dg) trafficking in neuronal cells. (**A**) Scheme of a potential Dg–exocyst interaction hypothesized based on the reported data on functions and subcellular localizations of Dg and exocyst complex proteins. The hypothesis predicts either Dg's function in mediating membrane-targeting of exocytic vesicles through interaction with the exocyst at the membrane site or exocyst-mediated Dg delivery to the cell membrane. Black circles outline proteins of the exocyst complex found to interact with Dg in the mass spectrometry screen. (**B**) Larval brain with GFP-negative *Dg* loss-of-function clones (outlined with yellow) immunostained with anti-Sec5. GFP and Sec5 are shown in separate channels. No obvious changes are observed in Sec5 protein levels or localization in *Dg* mutant clones when compared to GFP-positive control cells. Sec5 (red), GFP (green), and DAPI (blue). (**C**) GFP-negative *Sec15¹* mutant clones in larval brains show altered Dg localization when compared to neighboring control cells (GFP-positive). Dg (red), GFP (green), and DAPI (blue). (**D**) GFP-negative *Sec5^{E10}* mutant clones show an impaired Dg expression pattern in neuronal cells marked with the neuron-specific marker Elav. Dg (red), GFP (green), and Elav (blue). Elav, GFP, and Dg are also shown in separate channels. Yellow dashed line outlines *Dg⁻/Dg⁻* clonal area. (**E**) The surface of the larval brain showing control and *Sec15* clonal neuronal stem cells and their progeny. Note that in mutant cells (GFP-negative, yellow arrows) Dg protein is enriched in cytoplasmic puncta, more randomly distributed, and not properly delivered to the membrane in comparison to controls (green arrows). Dg (red), GFP (green), and Cad (blue). Scale bar 5 μm.
The online version of this article includes the following figure supplement(s) for figure 5:

**Figure supplement 1.** Dystroglycan (Dg) loss in neural cells does not have a significant effect on Sec5 protein.
**Figure supplement 2.** Perturbed exocyst causes downregulated dystroglycan (Dg) at the salivary gland cell membrane.
**Figure supplement 3.** Sec15-mediated trafficking shows protein specificity.

result in Dg downregulation or loss. Also, since Dg localization was perturbed in the brain and salivary gland, it suggests that the exocyst-mediated membrane trafficking of Dg is not neuron specific.

## Exocyst-mediated trafficking of Dg exhibits some specificity

The exocyst complex has been shown to be involved in polarized secretion as its deficiencies result in mislocalization of specific cell adhesion and signaling molecules in photoreceptor neurons and sensory organ precursors (*Jafar-Nejad et al., 2005*; *Mehta et al., 2005*). It was important to address whether the exocyst-dependent delivery of Dg in the brain is a general function in the secretory pathway or if it is Dg specific. Therefore, next, the distribution of various cell adhesion proteins in brains containing *sec15* mutant clones was studied. In particular, Discs large (Dlg1), Integrin PS2, Inflated (If), DE-Cadherin (Cad), its binding partner Armadillo (Arm), and a neural cell adhesion molecule (NCAM), Fasciclin II (FasII), were tested.

Discs large 1 (Dlg1) is a modular scaffolding protein that is expressed at specialized zones of the plasma membrane to regulate cell polarity through assembly of specific multiprotein complexes, which include signaling proteins, receptors, and ion channels (*Walch, 2013*). For example, the major ECM receptors, integrins, exist in a physical complex with the DLG scaffold in mammals (*Beumer et al., 2002*). It has been shown that, similar to Dg, integrins play important roles during brain morphogenesis, and defects in their functions result in the development of congenital muscular dystrophies (*Barraza-Flores et al., 2020*). The other factors critical for nervous system development are the cadherin family and the immunoglobulin-related superfamily of cell adhesion molecules, NCAMs (*Cammarota et al., 2020*; *Dumstrei et al., 2003*; *Fung et al., 2009*; *Neuert et al., 2020*). Cadherins connect cells with each other via adherens junctions, while NCAMs act as neuronal recognition molecules in the regulation of selective axon fasciculation. Surprisingly, it was found that in clones lacking one of the Sec proteins, localization of most of these cell adhesion molecules was not altered at the level of light microscopy (*Figure 5—figure supplement 3A–D*). The only protein that displayed aberrant localization in *sec15* clones was FasII. It aggregated in cytoplasmic foci in mutant cells, in contrast to the strict membrane localization in controls (*Figure 5—figure supplement 3E*, arrows). These results agree with previously published data that demonstrate the specific effect of the exocyst complex on the localization of various transmembrane cell adhesion and cell signaling proteins required for proper photoreceptor development (*Mehta et al., 2005*). One explanation for such specificity could be that both Dg and FasII proteins are glycosylated in the process of post-translation modification (*Dempsey et al., 2019*; *Nakamura et al., 2010*; *Parkinson et al., 2013*; *Patel et al., 1987*; *Snow et al., 1989*), while glycosylation has not been reported for any other tested here proteins. These data demonstrate that the function of the exocyst complex in Dg trafficking is rather specific and suggest that it might include transport of other proteins undergoing glycosylation, implying that the exocyst-dependent vesicular trafficking mechanism exists to spatiotemporally target a specific subset of cell adhesion molecules in neurons.

## Dg and exocyst expression levels and patterns correlate in the developing brain

During differentiation, axons grow out from the neuronal cell body, select the correct pathway for migration, choose specific target region within which they terminate, and recognize other cells to form synapses. All of these processes require robust protein trafficking within the neuron to result in the directional delivery of proteins to the axonal termini. Since the exocyst controls polarized secretion, and exocyst subunits have been implicated in neurite outgrowth and cell polarity (*Koon et al., 2018*; *Lira et al., 2019*), one possibility is that exocyst-mediated regulation of Dg trafficking takes place at specific stages, ensuring a temporally dynamic Dg protein expression pattern necessary for proper neuronal differentiation.

To understand Dg–exocyst relations in the brain, next, the expression patterns for Dg and Sec5 at different developmental stages were analyzed (*Figure 6—figure supplements 1* and *2*). In developing L3 larval brains, multiple Dg-positive puncta co-localize with Sec5-positive speckles, and this co-localization is enriched at the cell body periphery of differentiating neural cells (*Figure 6—figure supplement 1A, A'*, blue and yellow arrows) and in the regions containing axonal projections (*Figure 6—figure supplement 1B, B'*, yellow arrows). Similar to the dynamic Dg temporal expression,

during pupal stages, Sec5 protein is also observed in antennal lobe (AL) and CB neuropils (*Figure 6—figure supplement 2B*). In adult brains, expression of both proteins is strongly reduced compared to pre-adult stages (*Figure 6—figure supplement 2C*).

These data show that Sec5 has a temporal expression pattern reminiscent of Dg during development and a similar yet broader spatial pattern (*Figure 6—figure supplements 1* and *2*). This is consistent with involvement of Dg and the exocyst in a regulatory relationship. These data also suggest that exocyst–Dg regulations would take place predominantly in the developing brain, where the processes of neuronal and glial differentiation and neuropil formation occur.

## Exocyst-mediated Dg regulation is required for proper MB assembly

Next, to determine if exocyst-dependent Dg trafficking has a functional role in brain development, the assembly of the MB was investigated. This neuropil was particularly interesting for this research since the MB exhibits a temporal sequence in layer formation, in which younger neurons project first into the core and shift to the surrounding layers as they differentiate. In humans, to form the layered brain cortex, neurons migrate into different layers, and the migration process depends on the timing of when these neurons were born. *Drosophila* neurons do not migrate; however, MB cell bodies and their axonal projections are organized into layers, which also depends on the MB neuron birth timing (*Kurusu et al., 2002*; *Li et al., 2018*). The brain of dystroglycanopathy patients is characterized as cobblestone brain – a significant anomaly of cortical layering caused by neuronal overmigration. The expression data show that Dg tends to be expressed at higher levels in the youngest neurons, suggesting that the MB could be a good model to understand whether the exocyst and Dg functionally interact during brain compartmentalization and layer formation.

To test for functional interaction of the exocyst complex with Dg, the architecture of MBs of trans-heterozygous mutants with a reduced copy number of *Dg* and one of the exocyst complex components was analyzed. In particular, the exocyst components that were found in the mass spectrometry screen to directly interact with Dg as well as one additional exocyst component, Sec15, were studied (*Figure 5A*, red circles outlined in black). In addition, trans-heterozygous mutants with one functional copy of *Sec15* and *Sec5*, *Sec6*, or *Sec10* were examined.

Importantly, reduction by one copy of *Sec* components in a *Dg* heterozygous background results in a significant increase of both underdeveloped and misguided MB phenotypes when compared to just *Dg/+* heterozygous animals (*Figure 6A, B* and *Supplementary file 1*, see also *Figure 6—figure supplement 3*). The appearance and frequency of these phenotypes are similar to MB abnormalities observed in *Dg* loss-of-function mutants (*Figure 3A, B*). These data show that in the brain, the exocyst complex genetically interacts and cooperates with Dg to control proper brain compartmentalization. Furthermore, simultaneous reduction by one copy of two genes encoding exocyst components also causes MB phenotypes, which additionally confirms that exocyst-mediated trafficking plays a role in the assembly of MB lobes (*Figure 6A, B*, *Figure 6—figure supplement 3*, *Supplementary file 1*).

Together, the genetic interaction data support the findings from the mass spectrometry analysis about Dg–exocyst interaction and demonstrate that the exocyst complex plays a role in Dg transport in differentiating neurons (*Figure 6C*). These results also emphasize the importance of precise regulation of the expression and subcellular localization of Dg for the correct fasciculation of the MB lobes and brain compartmentalization.

## Discussion

These data show that the precise expression of the cell adhesion receptor Dg is critical for the proper establishment of brain neuropils in *Drosophila*. Dg loss and gain of function leads to abnormal brain compartmentalization, a phenotype similar to the anatomical brain defects of dystroglycanopathy patients. The analyses of the MB neuron-specific Dg up- and downregulation show that Dg is required for proper axonal pathfinding and neuropil assembly, and its abnormal expression in these neurons is sufficient to cause dramatic axonal misguidance and brain deformation phenotypes. It is crucial that Dg is only expressed at high levels in differentiating neurons, while in the mature neurons, Dg levels are notably lower. Since Dg establishes ECM composition and organization, which is key for proper fasciculation and establishment of functional neuropils that control various behaviors, these data propose that accurate subcellular localization of Dg is essential for proper

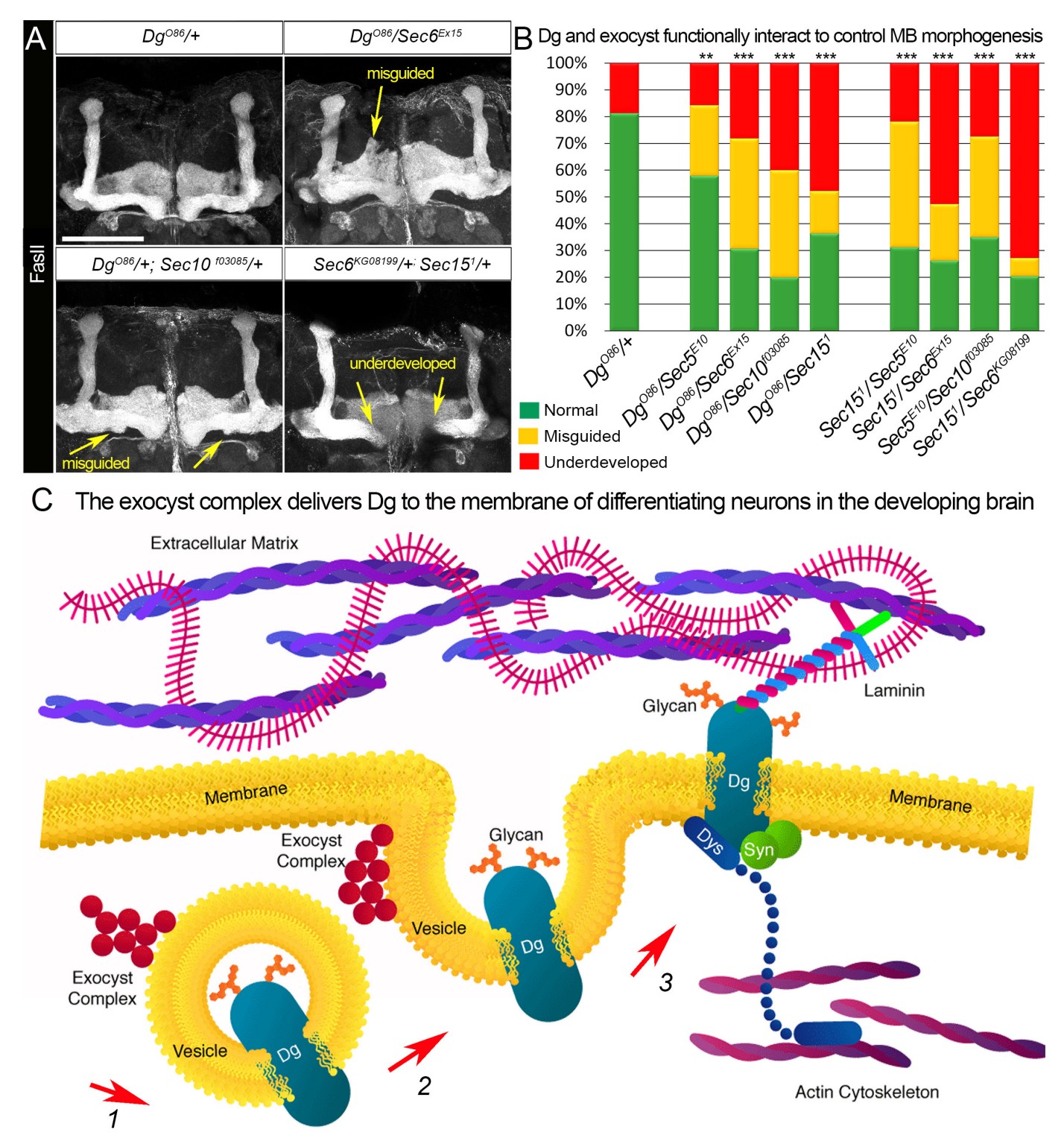

**Figure 6.** Proper patterning of the mushroom body (MB) neuropil is guaranteed by joint exocyst–dystroglycan (Dg) function. (**A**) Abnormal MB lobe architecture phenotypes are observed in trans-heterozygous animals carrying only one copy of *Dg* and one copy of *Sec* (*Dg^O86^/+; Sec10^f03085^/+* and *Dg^O86^/Sec6^Ex15^*), confirming a Dg–exocyst functional interaction in the process of MB morphogenesis (compare to *Figure 3G–H*). Viable combination of mutations in different exocyst subunits (*Sec6^KG08199^/+; Sec15^1^/+*) results in similar phenotypes. The MB lobes are marked with Fasciclin II (FasII). Scale bar 50 μm. For more phenotypes, see also *Figure 6—figure supplement 3*. (**B**) Bar graph presents the quantification of the phenotypes observed in the genetic interaction analysis (see also *Supplementary file 1*). Reduction by one copy of any two of the analyzed genes significantly affects MB

*Figure 6 continued on next page*

*Figure 6 continued*

morphogenesis. For comparison of MB phenotypes, $\chi^2$ test was used. ***p≤0.001; **p≤0.01. See also *Supplementary file 1*. (**C**) In differentiating neurons, the Dg protein is loaded into the exocyst-positive secretory vesicle where it may be glycosylated (1). Then, the vesicle transports the glycoprotein Dg to the membrane. Upon vesicle fusion with the membrane (2), Dg is localized at the membrane where it acts as the extracellular matrix receptor (3). Exocyst-mediated delivery of Dg is necessary for the establishment of brain compartments and proper neuronal networking.

The online version of this article includes the following figure supplement(s) for figure 6:

**Figure supplement 1.** Dystroglycan (Dg) co-localizes with the exocyst in the differentiating neurons.

**Figure supplement 2.** Dystroglycan (Dg) co-localizes with the exocyst in the developing brain.

**Figure supplement 3.** Mushroom body (MB) neuropil formation depends on exocyst–dystroglycan (Dg) interaction.

brain compartmentalization. To identify novel components that interact with Dg and influence the efficiency of its function in the nervous system, the Dg interactome in neurons was determined using mass spectrometry. Among the identified novel Dg interactors were several proteins of the exocyst complex. The data demonstrate that the exocyst mediates Dg trafficking to the plasma membrane of differentiating neurons, suggesting that Dg dynamics in the developing brain are at least partially governed by the exocyst complex (*Figure 6C*).

Currently, mass spectrometry experiments for protein complexes identification still encounter considerable limitations of data analysis. Firstly, Dg is a transmembrane protein, and affinity or immune purifications of transmembrane membrane proteins always pose a challenge as they must be extracted from membrane with their specifically interacting proteins. The success of such purification is dependent on the choice of detergent used. Here, the affinity purification specifically pulls down the exocyst complex. Strikingly, six of the eight components of the exocyst complex could be identified, demonstrating the specificity of the affinity purification. However, it cannot be ruled out that other interacting protein components might be lost, especially those that are transiently bound, due to the usage of detergent in the affinity purification that might disrupt transient or weak interactions. Therefore, alternative methods have recently been established that allow for monitoring protein–protein interaction of membrane-embedded proteins. These methods are called BioID or APEX proximity labeling approaches (*Roux et al., 2018*; *Yoo and Rhee, 2020*). However, such approaches would have required (i) fusion of Dg with the appropriate enzyme used in proximity labeling approaches and (ii) strictly quantitative analyses using stable isotopes to subtract the background of non-specific protein interactors.

Secondly, due to the lethality caused by pan-neuronal Dg overexpression, only Dg interactions that occur in adult, fully differentiated neurons could be detected in the Dg mass spectrometry screen. Since the data show that regulation of Dg expression is critical during brain development, it would be interesting to identify the neuronal type-specific and developmental time-specific Dg interactomes. In particular, it would be important to dissect MB lobe-specific interactors because MB neural progenitors generate different types of closely related neurons at specific times during an animal's development. For proper function, these neurons must cluster and synapse in a stereotyped fashion, which predominantly depends on selective cell adhesion (*Kucherenko and Shcherbata, 2013*; *Sanes and Zipursky, 2020*). Mammalian neural progenitors also produce multiple neuron types in the course of an individual's development to establish brain compartments and cortical layers, among which differential neuronal connections and functional neural circuits are assembled. This allows the processing of information, control of behavior, learning, memory, and plasticity of each individual. Interestingly, the mammalian cerebral cortex has a layered organization, which is composed of neurons born during different stages of development, as the deepest layers are formed by early-born neurons and the more superficial layers by late-born neurons, which migrate past the deep layers. Congenital MDs caused by Dg insufficiency are characterized by the cobblestone brain appearance, which is caused by neuron overmigration into the arachnoid space, resulting in cortical dysplasia and cortical layering defects (*Ackroyd et al., 2011*; *Combs and Ervasti, 2005*; *McDearmon et al., 2006*; *Satz et al., 2010*). The cause of this overmigration defect is impaired interactions between glia limitans and the ECM of the basement membrane (*Siegenthaler and Pleasure, 2011*; *Barkovich et al., 2012*). Moreover, it has been shown that Dg can directly interact with

secreted axon guidance cues and their transmembrane receptors, acting as a scaffold for extracellular axon guidance decisions (*Lindenmaier et al., 2019*; *Wright et al., 2012*). The most important domain required for this function is the sugar-decorated extracellular domain of Dg. Moreover, in mice, spatiotemporal persistence of functionally glycosylated Dg during the fetal stage could rescue severe cortical dysplasia, confirming that there is a temporal requirement for Dg glycosylation during brain development (*Sudo et al., 2018*). In general, the glycosylated cell surface binding receptors provide a physical link between the cytoskeleton and the ECM. Loss of glycosylation of these molecules contributes to functional defects during development by reducing binding to the ECM (reviewed in *Nickolls and Bönnemann, 2018*).

The process of protein glycosylation is a multistep enzymatic process occurring within distinct subcellular membrane-defined compartments, namely endoplasmic reticulum and Golgi. Generated at the Golgi apparatus, exocytic vesicles transport proteoglycans using cytoskeletal tracks and motor proteins to the plasma membrane to ensure that the glycosylated parts of the proteins do not face the cytoplasm (*Heider and Munson, 2012*). Exocytic vesicle fusion at the target membrane is facilitated by SNARE proteins present on both membranes. Dg is heavily glycosylated; therefore, it is logical to propose that exocytic vesicles would be involved in Dg trafficking from recycling endosomes to the plasma membrane (*Figure 6C*). Moreover, Dg contains an epitope called matriglycan, which is necessary to bind extracellular proteins, and Dg presence on the neuronal cell membrane is critical for the establishment of the ECM that allows proper nervous system compartmentalization. In particular, Dg is required for the organization of laminins and other ECM molecules in the basement membrane, which provides a permissive growth substrate for axons (*Clements and Wright, 2018*; *Lindenmaier et al., 2019*; *Wright et al., 2012*). Therefore, the delivery of Dg to the proper sites is of utmost importance, and these data show that the exocyst could regulate stage-dependent Dg expression pattern in the developing brain.

Recently, genetic studies show that partial loss-of-function variants of at least two components of human exocyst complex, EXOC7 and EXOC8, are associated with a recessively inherited disorder characterized by brain atrophy, seizures, and developmental delay, and in severe cases, microcephaly and infantile death (*Coulter et al., 2020*). Several major functions associated with exocyst-mediated vesicle trafficking have been described: polarized exocytosis, cell migration, tumor invasion, cytokinesis, autophagy, and ciliogenesis (reviewed in *Wu and Guo, 2015*). In differentiating neurons, the exocyst is recruited to sites of membrane expansion such as axonal growth cones, tips of neurites, and branching points. It was suggested that the exocyst enrichment at these points is because the fusion of exocyst vesicles with the cell membrane helps to increase the membrane surface needed for neuronal cell growth, path-finding, and synapse establishment (*Das et al., 2014*; *Lira et al., 2019*).

This study shows that in Dg and exocyst mutants MB lobes were dramatically underdeveloped and misguided. The exocyst helps to remodel the actin cytoskeleton during dynamic shape changes of the plasma membrane that occur, for example, during axonal growth (*Holly et al., 2015*; *Zhao et al., 2013*). Also, the DGC is involved in regulation of the actin cytoskeleton as it physically connects the actin cytoskeleton to the ECM. Therefore, interplay between these two important complexes could influence the process of axonal growth. However, a more interesting hypothesis is that the exocyst could be involved in Dg trafficking. During synaptogenesis, the exocyst has been shown to play a role in the localized delivery of several neurotransmitter receptors (*Gerges et al., 2006*; *Koon et al., 2018*; *Lee and Schwarz, 2016*; *Riefler et al., 2003*; *Sans et al., 2003*). For example, previous data show that Dg is required for proper localization of glutamate receptor at the *Drosophila* neuromuscular junction (*Marrone et al., 2011b*). In addition, the vertebrate data suggest that Dg is required for organization of the ECM at synapses, suggesting that Dg delivery to the tips of neurons could be important for the stabilization of neuronal connectivity. The data here show that the exocyst could be involved in polarized delivery of Dg necessary to establish functional synapses. However, it would be important to analyze in greater detail the distribution of Dg-dependent factors and synapse functionality upon exocyst malfunction. Moreover, as the data show that in the developing *Drosophila* brain the expression patterns of Dg and exocyst have temporal characteristics, it will be important to analyze whether the exocyst complex regulates Dg expression in vertebrates and whether this regulation in the developing mammalian brain is also developmental stage specific.

# Materials and methods

**Key resources table**

| Reagent type (species) or resource | Designation | Source or reference | Identifiers | Additional information |
|---|---|---|---|---|
| Antibody | Anti-Dg (rabbit polyclonal) | Gift from Hannele Ruohola-Baker (*Deng et al., 2003*) | Dg | IF(1:1000) |
| Antibody | Anti-Sec5 (mouse monoclonal) | Gift from Thomas Schwarz (*Langevin et al., 2005*) | Sec5 | IF(1:50) |
| Antibody | Anti-GFP (chicken polyclonal) | Abcam | Cat# ab, 13970 | IF(1:5000) |
| Antibody | Anti-FAsII (mouse monoclonal) | Developmental Studies Hybridoma Bank | Cat# 1D4 | IF(1:50) |
| Antibody | Anti-Dlg (mouse monoclonal) | Developmental Studies Hybridoma Bank | Cat# 4F3 | IF(1:20) |
| Antibody | Anti-Elav (mouse monoclonal) | Developmental Studies Hybridoma Bank | Cat# 9F8A9 | IF(1:20) |
| Antibody | Anti-Arm (mouse monoclonal) | Developmental Studies Hybridoma Bank | Cat# N2 7A1 | IF(1:20) |
| Antibody | Anti-integrin alphaPS2 (mouse monoclonal) | Developmental Studies Hybridoma Bank | Cat# CF.2C7 | IF(1:50) |
| Antibody | Anti-DE-Cad (rat monoclonal) | Developmental Studies Hybridoma Bank | Cat# DCAD2 | IF(1:50) |
| Antibody | Anti-mouse IG1 Cy3 (goat polyclonal) | Jackson ImmunoResearch | Cat# 115-165-205 | Secondary antibody IF(1:500) |
| Antibody | Anti-rabbit Alexa 568 (goat polyclonal) | Thermo Fisher Scientific | Cat# A-11011 | Secondary antibody IF(1:500) |
| Antibody | Anti-rabbit Alexa 488 (goat polyclonal) | Thermo Fisher Scientific | Cat# A-11039 | Secondary antibody IF(1:500) |
| Genetic reagent (*Drosophila melanogaster*) | *w[1118]* | Bloomington *Drosophila* Stock Center | BDSC: 5905 FBgn0003996 | Wild-type strain |
| Genetic reagent (*D. melanogaster*) | *Oregon-R-C* | Bloomington *Drosophila* Stock Center | BDSC: 5 FBgn0003996 | Wild-type strain |
| Genetic reagent (*D. melanogaster*) | *insc-Gal4* | Bloomington *Drosophila* Stock Center | BDSC: 8751 | *w[*]; P{w[+mW.hs]= GawB}insc[Mz1407]* |
| Genetic reagent (*D. melanogaster*) | *C305a-Gal4* | Bloomington *Drosophila* Stock Center | BDSC: 30829 | *w[*]; P{w[+mW.hs]= GawB}Cka[c305a]* |
| Genetic reagent (*D. melanogaster*) | *C309-Gal4* | Bloomington *Drosophila* Stock Center | BDSC: 6906 | *w[*]; P{w[+mW.hs]= GawB}c309* |
| Genetic reagent (*D. melanogaster*) | *201Y-Gal4* | Bloomington *Drosophila* Stock Center | BDSC: 4440 | *w[1118]; P{w[+mW.hs]= GawB}Tab2[201Y]* |
| Genetic reagent (*D. melanogaster*) | *elav-Gal4* | Bloomington *Drosophila* Stock Center | BDSC: 458 | *w[1118], elavGal4; tubGal80ts* (temperature sensitive) |
| Genetic reagent (*D. melanogaster*) | *FRT40A GFP* | Bloomington *Drosophila* Stock Center | BDSC: 5629 | *hsFlp; Ubi GFP FRT 40A/CyO* (clone induction line) |
| Genetic reagent (*D. melanogaster*) | *FRT G13 GFP* | Bloomington *Drosophila* Stock Center | BDSC: 5826 | *hsFlp; FRTG13 GFP/CyO* (clone induction line) |
| Genetic reagent (*D. melanogaster*) | *FRT 82B GFP* | Bloomington *Drosophila* Stock Center | BDSC: 5827 | *hsFlp; +; FRT 82B GFP/TM3* (clone induction line) |
| Genetic reagent (*D. melanogaster*) | *UAS-Dg::GFP* | Gift from Marie-Laure Parmentier (*Bogdanik et al., 2008*) | Dg | *UAS-Dg::GFP* (Dg tagged GFP under control of UAS promoter) |
| Genetic reagent (*D. melanogaster*) | *UAS-Dg* | Gift from Hannele Ruohola-Baker (*Deng et al., 2003*) | Dg | *UAS-Dg* (Dg gene under control of UAS promoter) |

*Continued on next page*

*Continued*

| Reagent type (species) or resource | Designation | Source or reference | Identifiers | Additional information |
|---|---|---|---|---|
| Genetic reagent (*D. melanogaster*) | *UAS-DgRNAi* | Gift from Hannele Ruohola-Baker (*Deng et al., 2003*) | Dg | *UAS-DgRNAi* (Dg RNAi construct under control of UAS promoter) |
| Genetic reagent (*D. melanogaster*) | *DgO86/CyO* | Gift from Robert Ray (*Christoforou et al., 2008*) | Dg | *Dg^O86^/CyO* (premature stop codon at 87 aa, strong hypomorph or null) |
| Genetic reagent (*D. melanogaster*) | *DgO55/CyO* | Gift from Robert Ray (*Christoforou et al., 2008*) | Dg | *Dg^O55^/CyO* (premature stop codon at 653 aa, strong hypomorph or null) |
| Genetic reagent (*D. melanogaster*) | *FRT G13 DgO55/CyO* | Gift from Robert Ray (*Christoforou et al., 2008*) | Dg | *w[\*]; FRT G13 Dg^O86^/SM6a* (line for Dg mutant clone induction line) |
| Genetic reagent (*D. melanogaster*) | *FRT40A, Sec5^E10^/CyO* | Gift from Yohanns Bellaiche (*Langevin et al., 2005*) | Sec5 | *w[\*]; FRT40A, Sec5^E10^/CyO* (null) |
| Genetic reagent (*D. melanogaster*) | *FRT82B Sec15^1^/TM3* | Gift from Yohanns Bellaiche (*Langevin et al., 2005*) | Sec15 | *w[\*]; FRT82B Sec15^1^/TM3* (premature stop codon, strong hypomorph or null) |
| Genetic reagent (*D. melanogaster*) | *FRT G13, Sec6^KG08199^/CyO* | Gift from Yohanns Bellaiche (*Langevin et al., 2005*) | Sec6 | *w[\*]; FRT G13, Sec6^KG08199^/CyO* (P-element insertion, null) |
| Genetic reagent (*D. melanogaster*) | *FRT 82B, Sec10^f03085^/TM6, Tb* | Gift from Yohanns Bellaiche (*Langevin et al., 2005*) | Sec10 | *FRT 82B, Sec10^f03085^/TM6, Tb* (PBac-element insertion, null) |
| Genetic reagent (*D. melanogaster*) | *FRT G13 Sec6^Ex15^/CyO* | Gift from Mark Metzstein (*Jones et al., 2014*) | Sec6 | *w[\*]; FRT G13 Sec6^Ex15^/CyO, Act-GFP* (null) |
| Software, algorithm | Adobe Photoshop | Adobe | Adobe CC | |
| Software, algorithm | Zen 2011 | Carl Zeiss | Zen 2011 | |
| Software, algorithm | MaxQuant software 1.3.0.5 | *Cox and Mann, 2008* | MaxQuant | |
| Software, algorithm | Markov clustering algorithm | https://micans.org/mcl/ | MLC | |
| Software, algorithm | Human disease-association enrichment analysis | http://ctdbase.org/tools | Disease Association | |
| Software, algorithm | Protein domain structure analysis | http://smart.embl-heidelberg.de | SMART | |
| Software, algorithm | Functional protein-association network clustering | https://string-db.org/ (*Szklarczyk et al., 2015*) | String v.10 | |
| Chemical compound, drug | Brilliant Blue R | Sigma Aldrich | Cat# 27816-25G | |
| Chemical compound, drug | Acetic acid | Sigma Aldrich | Cat# 27225-1 L-M | |
| Chemical compound, drug | Chloroform | Sigma Aldrich | Cat# 288306–2L | |
| Chemical compound, drug | Glycerol | Sigma Aldrich | Cat# G6279-1L | |
| Chemical compound, drug | Sodium azide | Sigma Aldrich | Cat# S2002-25G | |
| Chemical compound, drug | Formaldehyde, 16% | Polysciences Inc | Cat# 18814-20 | Methanol free, ultra pure |

*Continued on next page*

*Continued*

| Reagent type (species) or resource | Designation | Source or reference | Identifiers | Additional information |
|---|---|---|---|---|
| Commercial assay or kit | RealTime ready Cell Lysis Kit | Roche | Cat# 06366 821001 | |
| Commercial assay or kit | GFP-Trap A Kit | Chromotek | Cat# 5062685 | |
| Other | DAPI stain | Sigma Aldrich | Cat# D9542-10MG | IF concentration used: 1 µg/mL |
| Other | Normal Goat Serum | Abcam | Cat# ab7481 | |
| Other | Trans-Blot Turbo Mini PVDF Transfer Packs 0.2 µm | Bio-Rad | Cat# 1704156 | |
| Other | Immun-Blot PVDF/Filter Paper Sandwiches | Bio-Rad | Cat# 1620218 | |
| Other | Precision Plus Protein Kaleidoscope Prestained Protein Standard | Bio-Rad | Cat# 1610375 | |
| Other | 10× Tris/Glycine/SDS Running Buffer | Bio-Rad | Cat# 1610772 | |
| Other | NuPAGE Novex 4–12% Protein Gels | Thermo Fisher Scientific | Cat# NP0321PK2 | |
| Other | PicoFrit Columns | New Objective | Cat# PF360-75-15-N | |
| Other | Paraplast Plus | Sigma Aldrich | Cat# 76258-1KG | |
| Other | Casein Blocking Buffer 10x | Sigma Aldrich | Cat# B6429-500ML | |
| Other | Hematoxylin Solution, Mayer's | Sigma Aldrich | Cat# MHS16-500ML | |
| Other | Eosin Y solution, aqueous | Sigma Aldrich | Cat# HT110232 | |
| Other | DPX Mountant for histology | Sigma Aldrich | Cat# 06522-100ML | |
| Other | PBS buffer (10× Dulbecco's) | AppliChem | Cat# A0965,9010 | |
| Other | LSM700 confocal laser-scanning microscope | Carl Zeiss | LSM700 | |
| Other | Hyrax M25 microtome | Carl Zeiss | Hyrax M25 | |
| Other | ReproSil-Pur analytical column 120 C18-AQ | Dr. Maisch GmbH | ReproSil-Pur | |
| Other | Nanoflow liquid chromatography system EASY n-LC 1000 | Thermo Scientific | Nanoflow | |
| Other | Q Exactive Hybrid Quadrupole-Orbitrap | Thermo Scientific | Orbitrap | |

## Experimental model

*Drosophila* stocks and crosses were kept on standard corn-meal, yeast, glucose agar medium at 25°C under 12 hr:12 hr light:dark cycles.

To co-immunoprecipitate Dg from the brain tissue, transgenic animals bearing the C-terminally GFP-tagged Dg construct *UAS-Dg::GFP* (*Bogdanik et al., 2008*) were crossed to *elav-Gal4; Tub-Gal80^{ts}* (BDSC) driver line allowing for Dg expression in neurons. To avoid the high rate of lethality caused by Dg overexpression in neurons (*Yatsenko et al., 2014b*), crosses were kept at permissive temperature (18°C) during developmental stages, and as flies hatched from the pupal cases they were moved to restrictive temperature (29°C) for 5 days. After this, flies' heads were separated from the rest of the bodies and subjected to further procedures.

For neuronal upregulation of Dg, *insc-Gal4* (*Mz1407-Gal4*, BDSC 8751) was used. For overexpression or downregulation of Dg in the MB, the following drivers were used: *c305a-Gal4* (BDSC 30829), *c309-Gal4* (BDSC 6906), and *201Y-Gal4* (BDSC 4440) (*Aso et al., 2009*). Transgenic lines for manipulation of Dg expression levels, *UASt-Dg* and *UASt-dsDgRNAi* (*Deng et al., 2003*), were used.

To address Dg expression in exocyst mutants and Sec5 expression in *Dg* mutant, brain and salivary gland clones carrying mutations in different exocyst subunits were analyzed and compared to non-clonal neighboring (control) cells. To generate GFP-negative mutant clones, females of the genotypes *hs-Flp; Ubi-GFP FRT40A/CyO*, *hs-Flp; FRTG13 Ubi-GFP/CyO*, and *hs-Flp; +; FRT82B Ubi-GFP/TM3* carrying suitable FRT constructs were crossed with males of the genotypes *FRT40A*, *Sec5^{E10}/CyO*, *FRTG13 Sec6^{Ex15}/CyO*, *FRTG13 Sec6^{KG08199}/CyO*, and *FRT82B Sec15^1/TM3* carrying mutations in exocyst complex subunits and respective FRT sites. For Dg mutant clone generation, *hs-Flp; FRTG13 Ubi-GFP/CyO* females were crossed with *FRTG13 Dg^{O86}/CyO* males.

In genotypes carrying expressing flippase under control of the *heat-shock* gene promoter (*hs-Flp*), clone generation was induced by exposing 1st–2nd instar larval progeny to 1 hr heat shocks (37°C) for two consecutive days. Brain and salivary gland clones carrying mutations in either exocyst complex components or Dg were analyzed at L3 larval, pupal, and adult developmental stages. Note that the exocyst mutant clones in brain did not survive until adulthood.

## Co-immunoprecipitation and western blot analysis

Whole lysates for general co-immunoprecipitation were prepared from heads of approximately 1-week-old flies. Tissue was homogenized with VWRR Disposable Pellet Mixers and lysed in RIPA buffer containing 50 mM Tris-HCl (pH 7.5), 125 mM NaCl, 5% glycerol, 0.5% NP40, 0.25% Na-deoxycholate, 1.5 mM MgCl$_2$, 1 mM dithiothreitol, 25 mM NaF, 1 mM Na$_3$VO$_4$, 1 mM EDTA, 2 mM EGTA, and protease inhibitors. Samples were then centrifuged at 15,000 *g* for 15 min at 4°C, and then 1.3 mg of supernatants were immunoprecipitated with GFP-Trap beads coupled with anti-GFP antibody (ChromoTek) following the manufacturer's instructions. Four percent of total protein extracts used for immunoprecipitation was loaded as input.

## Coomassie Brilliant Blue (CBB) staining

The CBB staining was performed as previously described (*Lawrence and Besir, 2009*). In brief, CBB G-250 was dissolved in double-distilled water in a concentration of 60–80 mg/L and 35 mM HCl was added as the only other compound in the staining solution. The gel from SDS-PAGE was rinsed with double-distilled water and incubated in CBB staining solution overnight at room temperature with gentle shaking. Next, the stained gel was de-stained through washing with double-distilled water.

## Gel electrophoresis, in-gel digestion, and mass spectrometry (LC-MS/MS)

Proteins were separated by one-dimensional SDS-PAGE (4–12% NuPAGE Bis-Tris Gel, Invitrogen) and stained with Coomassie Blue G-250 (Sigma). The complete gel lanes were cut into 23 equally sized slices. Proteins were digested as described previously (*Shevchenko et al., 2006*). Briefly, proteins were reduced with 10 mM DTT for 50 min at 50°C, then alkylated with 55 mM iodoacetamide for 20 min at 26°C. In-gel digestion was performed with Lys-C (Roche Applied Science) overnight. Extracted peptides from gel slices were loaded onto the in-house packed C18 trap column (ReproSil-Pur 120 C18-AQ, 5 µm, Dr. Maisch GmbH; 20 × 0.100 mm) at a flow rate of 5 µL/min loading buffer (2% acetonitrile, 0.1% formic acid). Peptides were separated on the analytical column (ReproSil-Pur 120 C18-AQ, 3 µm, Dr. Maisch GmbH; 200 × 0.050 mm, packed in-house into a PF360-75-15-N PicoFrit capillary, New Objective) with a 90 min linear gradient from 5% to 40% acetonitrile containing 0.1% formic acid at a flow rate of 300 nL/min using a nanoflow liquid chromatography system (EASY n-LC 1000 Thermo Scientific) coupled to hybrid quadrupole-Orbitrap (Q Exactive, Thermo Scientific). The mass spectrometer was operated in data-dependent acquisition mode where survey scans acquired from m/z 350–1600 in the Orbitrap at resolution settings of 70,000 FWHM at m/z 200 at a target value of 1 × 10e6. Up to 15 most abundant precursor ions with charge states 2+ or more were sequentially isolated and fragmented with higher collision-induced dissociation (HCD) with normalized collision energy of 28. Dynamic exclusion was set to 18 s to avoid repeating the sequencing of the peptides.

## Mass spectrometry data analysis

The generated raw mass spectrometry files were analyzed with MaxQuant software (*Cox and Mann, 2008*) (version 1.3.0.5, using Andromeda search engine) against UniProtKB *Drosophila melanogaster*

database containing 18,826 entries (downloaded in April 2013) supplemented with common contaminants and concatenated with the reverse sequences of all entries. The following Andromeda search parameters were set: carbamidomethylation of cysteines as a fixed modification, oxidation of methionine and N-terminal acetylation as a variable modification, and Lys-C specificity with no proline restriction and up to two missed cleavages. The MS survey scan mass tolerance was 7 ppm and for MS/MS 20 ppm. For protein identification, minimum of five amino acids per identified peptide and at least one peptide per protein group were required. The false discovery rate was set to 1% at both peptide and protein levels. 'Re-quantify' was enabled, and 'keep low scoring versions of identified peptides' was disabled. A twofold increase in any protein in *Dg::GFP* sample in comparison to control obtained from two independent biological replicates was considered as Dg-interacting protein.

## Immunohistochemistry

Larval, pupal, and adult brains were rapidly dissected in PBS and fixed in 4% formaldehyde (Polysciences, Inc), larval for 15 min and pupal-adult for 30 min. Staining was performed as described (*Kucherenko et al., 2012*). Samples were mounted in 70% glycerol. The following antibodies were used: polyclonal rabbit anti-Dg 1:1000 (*Deng et al., 2003*), polyclonal chicken anti-GFP 1:2000 (Invitrogen), monoclonal mouse anti-Sec5 1:50 (gift from Thomas Schwarz [*Langevin et al., 2005*]), and anti-FasII 1:20, anti-Dlg 1:20, anti-Elav 1:20, anti-Arm 1:50, anti-FasII 1:50, anti-αPS2 1:50, and rat anti-DE-cadherin 1:50 from Developmental Studies Hybridoma Bank. Alexa 488, 568, goat, anti-rabbit, and anti-chicken 1:500 (Molecular Probes). To visualize nuclei, a 10-min-long incubation with 1× DAPI (Sigma Aldrich) in PBS was performed.

## Histology of *Drosophila* brains

For analysis of adult brain morphology, 7 μm paraffin-embedded sections were cut from fly heads. To prepare *Drosophila* brain sections, the fly heads were immobilized in collars in the required orientation and fixed in Carnoy fixative solution (6:3:1 = ethanol:chloroform:acetic acid) at 4 °C overnight. Tissue dehydration and embedding in paraffin was performed as described previously (*Kucherenko et al., 2010*). Histological sections were prepared using a Hyrax M25 (Zeiss) microtome and stained with hematoxylin and eosin as described previously (*Shcherbata et al., 2007*). All chemicals for these procedures were obtained from Sigma Aldrich.

## Bioinformatic analyses

To arrange identified interactors in the functional protein-association network, STRING v10 database was used (*Szklarczyk et al., 2015*), with medium confidence score (0.04), and prediction methods that included neighborhood, gene fusion, co-occurrence, co-expression, experiments, databases, and text mining. Network is presented in a 'confidence' view, where thickness of lines connecting nodes represents confidence of association. The interaction scores of high confidence (0.700) were considered. Lines connect components clustered by MCL into protein complexes using the inflation parameter (3). Dashed lines show associations between components that do not form protein complexes. To assign protein cellular localization and molecular function and find human orthologs, the FB2016_02 release from FlyBase was applied. To search for human disease associations, the sites http://www.flyrnai.org (*Hu et al., 2011*) and http://www.genecards.org were used. For human disease-association enrichment analysis, the entry of 118 human genes (orthologs of identified Dg-interacting components) was examined with http://ctdbase.org/tools (Disease) tool. If p<0.001, the disease enrichment was considered as significant.

## Microscopy and image analyses

Images were obtained using a Zeiss LSM700 confocal laser-scanning microscope. Protein expression patterns and protein co-localizations were analyzed from confocal images taken in a z-stacks (1 μm step). Images were processed with ZEN Lite and Adobe Photoshop software. All parameters characterizing brain structure such as neuropils' area, length, and width were measured with tools of ZEN 2011 software.

To analyze the frequency of abnormal MB phenotypes, Z-stack confocal images of the entire adult brain with 1 μm intervals were captured. MBs were identified by FasII expression. The numbers of underdeveloped and misguided MB lobes in different mutants were quantified. All experiments

were performed at least in three biological replicates for each mutant fly line. The mean and standard deviation of the control and the mutant were calculated. For comparison of the observed phenotypes, two-way tables and $\chi^2$ test were used.

To analyze the intensity of Dg staining at the membranes of salivary gland cells, ImageJ software was used. The intensity of fluorescence was measured in the same size rectangular regions of interest between the membranes of two GFP-positive (control) and two GFP-negative (mutant) cells. Then, the average intensities were calculated. For comparison of the obtained intensities, two-tailed Student's t-test was used.

## Acknowledgements

We thank Hannele Ruohola-Baker, Joseph Matthew Bateman, Mark Metzstein, Yohanns Bellaiche, and Thomas Schwarz for flies and antibodies; Travis Carney for helpful discussion and comments on the manuscript; Omer Cicek for help with initial quantifications of MB phenotypes; Marko Shcherbatyy for drawing schemes; and all Shcherbata lab members for critical reading of the manuscript and helpful suggestions. This work was funded by the Hannover Medical School, Max Planck Society, VolkswagenStiftung (AZN3008, AZ97750), Behrens-Weise-Foundation, and EMBO YIP.

## Additional information

### Funding

| Funder | Grant reference number | Author |
| --- | --- | --- |
| EMBO | YIP | Halyna R Shcherbata |
| VolkswagenStiftung | AZN3008 | Halyna R Shcherbata |
| VolkswagenStiftung | AZ97750 | Halyna R Shcherbata |
| Behrens-Weise-Foundation | | Halyna R Shcherbata |

The funders had no role in study design, data collection and interpretation, or the decision to submit the work for publication.

### Author contributions

Andriy S Yatsenko, Formal analysis, Validation, Investigation, Visualization, Methodology; Mariya M Kucherenko, Formal analysis, Investigation, Visualization, Methodology, Writing - original draft; Yuanbin Xie, Investigation, Methodology; Henning Urlaub, Resources, Investigation, Methodology; Halyna R Shcherbata, Conceptualization, Resources, Data curation, Formal analysis, Supervision, Funding acquisition, Validation, Investigation, Visualization, Methodology, Writing - original draft, Project administration, Writing - review and editing

### Author ORCIDs

Halyna R Shcherbata  https://orcid.org/0000-0002-3855-0345

### Decision letter and Author response

Decision letter https://doi.org/10.7554/eLife.63868.sa1
Author response https://doi.org/10.7554/eLife.63868.sa2

## Additional files

### Supplementary files

• Supplementary file 1. Mushroom body (MB) morphology is affected by deregulation of dystroglycan (Dg) and the exocyst. **\*Misguided** = fused β lobes or β lobe neurons projecting into γ lobe space.**\*\*Underdeveloped** = smaller α or β lobes due to α lobe neurons projecting into β lobe space or vice versa.[a]Compared to *Oregon (Control)*.[b]Compared to *Dg$^{O86}$/+ (Control)*.[c]Compared to *Dg$^{O86}$/Sec6$^{Ex15}$*.[d]Compared to *Dg$^{O86}$/Sec10$^{f03085}$*.[e]Compared to *Dg$^{O86}$/Sec15$^1$*.For comparison of the observed phenotypes, two-way tables and $\chi^2$ test were used.

- Supplementary file 2. Dystroglycan (Dg)-interacting proteins in neurons.

- Supplementary file 3. Human disease enrichment in the dystroglycan (Dg) neuronal interactome network. For human disease-association enrichment analysis, the entry of human orthologs of identified Dg-interacting components (Table S2) was examined with Comparative Toxicogenomics Database (CTD) Disease Tool http://ctdbase.org/tools. The corrected threshold value of p<0.001 was used.

- Supplementary file 4. Functional enrichments in the dystroglycan (Dg) neuronal interactome network based on cellular component (GO) terms.

- Supplementary file 5. Dystroglycan (Dg)-associated components placed into the protein-interaction network identified by Markov clustering algorithm (MCL). STRING database (https://string-db.org) was used to identify functional networks with the interaction score of high confidence (0.700). Interacting protein are clustered by the MCL with the inflation parameter (3). For the schematic representation of the obtained interaction network, see *Figure 4D*.

- Transparent reporting form

## Data availability

The authors declare that all data supporting the findings of this study are available within the article and its supplementary information files or found on the Dryad Digital Repository (Data DOI: https://doi.org/10.5061/dryad.8sf7m0cmf).

The following dataset was generated:

| Author(s) | Year | Dataset title | Dataset URL | Database and Identifier |
|---|---|---|---|---|
| Yatsenko AS, Kucherenko MM, Xie Y, Urlaub H, Shcherbata HR | 2021 | Neuronal Dystroglycan Interactome | http://dx.doi.org/10.5061/dryad.8sf7m0cmf | Dryad Digital Repository, 10.5061/dryad.8sf7m0cmf |

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
