## [Decision Letter]

[Editors' note: this paper was reviewed by Review Commons.]

**Acceptance summary:**

This study combines genetics, imaging, and mass spec approaches to reveal the spatial dynamics and functional roles for the ECM receptor Dystroglycan during the formation of neuropils in the developing *Drosophila* brain. It identifies a large number of candidate Dg interactors and functionally demonstrates that one of these, the exocyst complex is essential for the function of Dg in neuropils. The *Drosophila* neuropil is analogous to the mammalian hippocampus, and parallels between these neuropil phenotypes and certain human dystrogycanopathies may suggest underlying mechanistic parallels.

**Decision letter after peer review:**

Thank you for submitting your article "Exocyst-mediated membrane trafficking of the ECM receptor Dystroglycan is required for proper brain compartmentalization" for consideration by *eLife*. Your article has been reviewed by three peer reviewers, and the evaluation has been overseen by a Reviewing Editor and K VijayRaghavan as the Senior Editor. The reviewers have opted to remain anonymous.

The reviewers have discussed the reviews with one another and the Reviewing Editor has drafted this decision to help you prepare a revised submission.

Summary:

This study combines genetic, imaging, and mass spec approaches to reveal a role for the ECM receptor Dystroglycan in the formation of neuropils during development of the *Drosophila* brain and implicates the exocyst complex as essential for this role. The *Drosophila* neuropil is analogous to the mammalian hippocampus, and parallels between these neuropil phenotypes and certain human dystrogycanopathies may suggest underlying mechanistic parallels.

Reviewers highlighted numerous strengths of the study: The dynamics of Dg localization in neuropil development are provocative and interesting. The use of mass spec to identify a large number of additional Dg candidate interactors is a valuable contribution. The use of genetic approaches to examine one class of interactors, the exocyst complex, provides functional validation. Finally, the disease-like phenotype associated with Dg perturbation is provocative and broadens the overall interest of the work.

Essential revisions:

1) Expand discussion of mass-spec interactome, including limitations of the analyses. As pointed out by one reviewer, the mass-spec results would be strengthened by biochemical evidence for interaction between Dg and the proteins identified by mass-spec (such as co-IP using antibodies specific for Dg interactors). The study nonetheless does present extensive genetic analyses for one group of interactors, components of the exocyst complex, and in my evaluation these genetic interaction experiments provide sufficient functional evidence of a regulatory role for the overall point of this work. What is lacking, however, is discussion of the other groups that were identified. Rather than perform new experiments, the authors should explicitly discuss the elements of protein interaction network presented in 4D, including any known links to Dg or Dg-related disease and limitations/caveats of the mass-spec approach. Such a discussion will provide the reader with valuable context about the Dg interactome.

2) Clarify some aspects of the exocyst-Dg interaction. The authors present data showing intracellular accumulation of Dg in exocyst mutant cells from salivary glands as well as various cell types in the brain, which demonstrates that exocyst regulation of Dg is not neuron-specific. Unadressed, however, is the key question of whether exocyst affects Dg through its general function in the secretory pathway or through a Dg-specific mechanism. Can the authors provide evidence one way or another, for instance by immunostaining sec mutants for other cell-surface ECM receptors or transmembrane proteins, or (given *eLife*'s COVID response), provide an argument for a general or specific scenario?

A second issue is that, in Figure 5C and 5D, it is difficult to distinguish cell-surface and intracellular Dg in the sec-mutant cells in larval brain. These data are central to the author's model that failed delivery of Dg to the surface of these cells leads to neuropil defects, but the small size and minimal cytoplasmic volume makes it hard for me to interpret these data on their own merits. Can the authors provide additional support for the cellular localization of Dg in larval brains-perhaps, for instance, by performing immunostaining of unpermeabilized tissue, or by providing other experimental evidence or (given *eLife*'s COVID response) scientific rationales for the localization of Dg in these cells?

---

## [Author Response]

Reviewer #1 (Evidence, reproducibility and clarity (Required)):Dystroglycan (Dg) is a conserved TM protein which connects and signals between the ECM and the intracellular actin cytoskeleton and is here established as a disease model for a form oflyssencephaly. Yatsenko et al. first demonstrate that in the brain, Dg staining appears and cedes with differentiation of neuropils, suggesting a function in neuropil formation and thus brain compartmentalization. Indeed, adult mutant escapers or brains of flies with neural overexpression of Dg show malformed neuropils such as the antennal lobe and mushroom body (MB), where some axons even cross the midline. Pleiotropic defects are clearly significant for α/β lobes of the MB, phenocopied by RNAi, and even stronger upon Dg overexpression, showing that proper levels/timing of Dg are critical for axon/brain development. Sub-class specific neuronal alteration of Dg confirms a requirement of Dg in MB neurons, but also suggests a broader/stronger effect on the MB if misexpression occurs in all neurons. The commonalities of these phonotypes with human dystroglycanopathy patients demonstrate the utility of the Drosophila model.Adult specific expression of Dg led to the identification of many potential Dg interactors with 6 of the 8 exocyst components being prominent and thus used for in vivo follow-up. Yatsenko et al. show that Dg is mislocalized (see below) in mutants of two Sec components and, importantly, they also nicely show genetic interactions in compound heterozygous animals. Similarly, transheterozygotes of Sec components show MB defects too, indicating that the exocyst has a role in the differentiation of the MB.

We would like to sincerely thank the reviewer for the positive evaluation of our work, careful reading of our manuscript, and helpful suggestions. We completely agree with all your critiques and comments. We have introduced all proposed changes to the revised version of our manuscript.

Major Points:Neuron specific interactome: the term “specific” is not adequate, as no comparison with other tissues or cell types shows a restriction of these interactors to neurons. Interactome in neurons would be more appropriate.

We agree, it is not “neuron-specific” because we have not compared it with other tissues, and actually some of the factors do interact with Dg in other cells (for example, Kibra in muscles). Therefore, we have changed “neuron-specific Dg interactome” to “Dg interactome in neurons”.

Also, the authors fail to comment on whether known interactors of Dg were identified, which would increase the reliability of the data.

Currently, few interactors have been identified for Dg in any organism, especially in the nervous system. Apart from several components of the Dystrophin Glycoprotein Complex per se such as Dystrophin, its homolog utrophin, and sarcoglycans, only a few ECM proteins, for example, agrin, eyes shut, pikachurin, perlecan, and laminins have been shown to bind to Dg. Encouragingly, we could also detect Eys, a *Drosophila* homolog of agrin and eyes shut. Interestingly, Dg interaction with agrins modulates the assembly of synapses (Bassat et al., 2017; Fallon and Hall, 1994; Hilgenberg et al., 2009; Liu et al., 2020), and apart from agrin, we have also identified several proteins playing a role in synaptic signaling. As mentioned above, recently Dg has been shown to interact with Kibra in both vertebrates and invertebrates (Iyer et al., 2019; Morikawa et al., 2017; Vita et al., 2018; Yatsenko et al., 2020). We have also detected Kibra in our screen, suggesting that a Dg-Hippo signaling interaction might be also important in the nervous system. As the reviewer suggested, we have added this information supporting our screen.

Judging by the methods, GFP-Dg expression was induced in 1 week old adults to identify Dg interactors in neurons. It is unclear why this time-point is chosen (which is past the strong expression shown in Figure 1). Key functional interactors for Dg function may have been missed using this approach. The discussion somewhat alludes to the issue, but does not state why earlier stages were not used.

We agree with the reviewer that in our experimental setup, we could have missed some of the Dg interactors. However, Dg overexpression with neuronal drivers during earlier stages is lethal. Therefore, we had to induce Dg overexpression in adults. We kept them for 5 days at 29°C to ensure the sufficient expression of the tagged protein for mass spectrometry. We explained this already in the Results sections describing the settings for the mass spec screen.

Figure 5/Figure 5—figure supplement 2: The authors describe that Sec mutants “dysregulate Dg localization”. Rather describe the changes. In the brain, there appears less membrane recruitment of Dg, but possibly higher intracellular puncta (speaking for their model). However, in the SG the latter is less clear. Why are homozygous WT tissues outlined separately? Please comment whether there is a difference between het and homozygous WT tissue (as levels of Dg important for Dg phenotypes). In the SG, it should be easy to quantify differences in staining intensity at WT:WT , WT:Mut, and Mut:Mut membranes which would make the data stronger (and show a greyscale image without the clonal border to judge the situation in which a WT cell abuts a mutant cell).

We thank the reviewer for this comment. We now describe the changes in Dg localization observed in neuronal Sec clones. The reviewer is correct: since there are no obvious differences in Dg localization in wt vs. heterozygous clones, we should not have outlined these cell populations separately. Moreover, in the salivary gland the difference between wt and heterozygous cells is not detectable. The marking of clone borders in Figure 5 has been adjusted.

As correctly noted by the reviewer, while the reduction of Dg delivery to the cell membrane is obvious in both tissues, subcellular Dg puncta are observed in brains, but not salivary gland cells. One possible explanation could be that neurons have very large surface area:volume ratios, requiring a great deal of membrane material to be synthesized, while salivary gland cells have a much smaller surface:volume ratio. Therefore, it could be that the expression levels of membrane proteins (including Dg) are higher in neurons than in salivary gland cells. Also, it is possible that the amount of the exocyst-dependent vesicle trafficking is different in these two cell types, *etc.*

As the reviewer suggested, we have now quantified differences in Dg staining intensity in wild type and mutant salivary gland cell membranes. This analysis shows that loss of one of the Sec proteins leads to the significant decrease in Dg presence on the cell membranes (60-70%). We thank the reviewer for this suggestion, since this new experiment allowed us to show the effect of the exocyst complex deficiencies on Dg localization in a quantitative manner.

We also changed Figure 5—figure supplement S2 as proposed, with no yellow outline in Dg channel plus GFP channel, and we feel that this has much improved the figure.

Discussion (and results): the authors state that Dg acts in a cell autonomous manner, although their experiments do not address this. All they can say is that affected neural subpopulations are sufficient to show (partial) phenotypes. No genotype/phenotype at the single cell level was performed.

True, now we have corrected this statement in the text.

Minor:Overly long in parts (in particular in the Introduction and Discussion).

We have tried to shorten these sections.

SG images not really higher resolution, just larger cells, thus easier to see. Along these lines, the scale bar should be moved to the edge of panel A.

Thank you, it is now corrected.

Figure 3: Indicate n values for quantifications in the figures (e.g. Dg/ Sec interactions). In the end, I found them in Supplementary file 1 after consulting the legend of Figure 3.

Thank you for mentioning this. We had completely neglected to include reference to Figure 3I. Now we have added the reference to the Results, saying (Figure 3I, for quantifications see Supplementary file 1). If we add all numbers to the figure, there would be no point of keeping the table; however, the table contains much additional important information, so we decided to keep the table and refer to it in the Results section and in the figure legend.

Stars in Figure 5A are difficult to see. Better add them to Figure 4.

We agree with the reviewer that it is difficult to see these stars. We cannot put stars only in 4D and exclude them from 5A, because 4D shows only components that we detected in the mass spec screen, while 5A contains all components of the exocyst complex with stars marking the components that were also detected in the screen. Now in 5A, we have outlined the components of the exocyst complex that were detected in the screen. Be hope that these lines are easier to see.

How were the alleles of the exocyst components verified?

We used only previously published exocyst component mutants, which we obtained from the labs of Yohanns Bellaiche and Mark Metzstein, where the alleles were generated and verified. All of them are considered to be null mutants. Since it is important for our genetic interaction studies, we have added the information on the nature of these mutants to our Key Resources Table.

“Complexome” is a horrible word creation!

Agree, deleted.

“..DG-exocyst interaction has a temporal expression pattern…” Weird sentence. How would an interaction have an expression pattern?

Thank you, corrected.

Reviewer #1 (Significance (Required)):Overall, this is well written paper interesting for a broad readership showing that Dg is expressed in a spatially and temporally dynamic pattern in forming neuropils. The work makes a strong and novel biochemical and genetic in vivo connection between the exocyst machinery and Dg providing substantial data in support of the exocyst being required for Dg delivery to the plasma membrane and being functionally required for neuropil/brain development in flies. Based on the conservation of these molecules, this work thus has important implications for mammalian development.Reviewers cross-commentingTo a large extent, I agree with the other reviewer. An independent way of showing an interaction would have been nice, although in a MS approach interactions are not necessarily direct. Therefore, the functional/genetic interaction is key and was nicely shown. If the authors have evidence for indirect interactions, they should discuss it. In addition, compound het interactions usually are very clear signs of functional relevance and again, were seen with different components of the Exorcyst complex.On the other hand, I am not sure why the other reviewer doubts about the Dg alleles. Some phenotypes are shown with both and the transhet combination is likely of intermediate strength. Perhaps I misunderstand the concerns.Reviewer #2 (Evidence, reproducibility and clarity (Required)):Summary:This is a well-written manuscript. Yatsenko et al. performed a very comprehensive spatio-temporal analysis of Dg expression. By means of mutational analysis, the authors showed that Dg is required for neuropil development. By means of immunoprecipitation-coupled mass spectrometry, proteins that associated with EGFP-tagged Dg were isolated. Among the different protein interacting partners, the authors investigated the roles of individual exocyst complex components in Dg trafficking and their involvement in neuropil development.

We thank the reviewer for kind words and for the effort in reviewing the manuscript. We have introduced the proposed changes to the revised version of our manuscript.

Major comments:The DgO86 is likely to be a null because of its premature stop (R87term), Dg055 produces 2/3 of the full-length protein. It is therefore not certain what would be the mutational effect the DgO55/DgO86 mutant presents.

The reviewer is correct, *Dg^O86^* is a null mutant and *Dg^O55^* produces a part of the full-length protein. In the original paper that describes generation and characterization of theses Dg alleles (Christoforou et al., 2008), it says that “*Dg^O86^* is associated with a C to T substitution at position 880 in Exon 3 that changes R87 to a STOP, thus truncating the protein within the N-terminal domain of α-Dystroglycan. *Dg^O55^* is an internal 23 bp deletion of the parent sequence that is replaced with a 3 bp insertion of unrelated sequence. This deletion is contained within Exon 11 and results in a frame shift that introduces a termination codon three amino acids after L653. Both *Dg^O86^* and *Dg^O55^* alleles have similar viability in hemizygous state and cause similar wing vein phenotypes with the same frequency”.

Therefore, both mutations behave like null mutants. We also have used and characterized these alleles previously and showed that both mutations lead to the appearance of similar muscle and brain phenotypes (Kucherenko et al., 2011; Kucherenko et al., 2008; Marrone et al., 2011a; Marrone et al., 2011b; Yatsenko et al., 2020). The reason we used an additional *Dg^055^* mutant allele is that the combination of *Dg^O86^/Dg^O86^*is lethal with very few escapers (less than 10%). Therefore, to increase the viability and to avoid any possible second-site mutations, we crossed these two different *Dg* allelic mutants. Importantly, our data show that *Dg^O86^/Dg^O86^* and *Dg^O86^/Dg^O55^* have similar brain phenotypes that occur with similar frequencies (see Figure 3I). We have now added the information about the nature of *Dg* alleles to our Key Resources Table.

No confirmatory experiments were presented to validate protein interacting partners identified from the mass spectrometric analysis.

As correctly noted by the reviewer, in this manuscript we did not intend to show direct biochemical interactions between Dg and identified candidates. Our mass spec analysis allowed us to detect larger multiprotein assemblies in which the components do not have to be directly associated with each other. We think that the transmembrane protein Dg acts as a scaffold that brings together different signaling components to the membrane. This idea has been supported by various previous studies, for example studies that show that Dg acts as a signaling hub in promoting NOS-HDAC, Hippo, and insulin signaling pathways (Cacchiarelli et al., 2010; Eid Mutlak et al., 2020; Vita et al., 2018; Watt et al., 2015; Yatsenko et al., 2020; Yatsenko et al., 2014). Also, Dg has been shown to be required for proper localization of several synaptic proteins (Bogdanik et al., 2008; Fallon and Hall, 1994; Marrone et al., 2011c; Wairkar et al., 2008). In addition to signaling factors, our mass spec analysis allowed us to identify components of the larger complexes that might play a role in Dg processing, such as Sec proteins. We’ve made sure that this information is clearly stated in the manuscript.

The double-heterozygotes experimental results are interesting. The different Sec mutants exert various degrees of "synthetic" mutant phenotypes when put on DgO86 heterozygous background. What are the nature of the different Sec mutants used in this experiments? Are they all null or hypomorphs? How are the effects of the sec mutants compared to DgO86/DgO55? It would be desirable if additional mutant alleles can be used to validate the findings.

As the reviewer suggested, we added the information on the nature of different Sec mutants used in our study (Key Resources Table). All analyzed Sec alleles appear to be null or strong hypomorphs. We have now compared these phenotypes not only to the control (*Dg/+*), but also to each other. Statistical analysis shows that even though there are some variations between different *Sec* alleles, they are not significant. We thank the reviewer for mentioning this; the new data are now included in Supplementary file 1, and they support even more our hypothesis that the entire exocyst complex plays a role in the membrane delivery of Dg.

Can the overexpression of individual Sec rescue Dg hypomorphic phenotype?

We think we misunderstood the reviewer’s question, because we are not sure what would be the logic of this experiment. Based on the published data, it is very unlikely that overexpression of a single Sec can promote the entire exocyst complex function. Also, Dg mutant alleles that we use have stop codons resulting in the truncation of the protein before the transmembrane domain. Since exocystpositive vesicles transport transmembrane proteins, it is implausible that the truncated Dg would be at all included in these vesicles. Even if overexpressing Sec could succeed in bringing more Dg to the membrane, it would only be delivering truncated Dg protein, which would not be expected to rescue any *Dg* mutant phenotypes. Therefore, the overexpression of individual Sec proteins would be unable to rescue loss of membrane Dg.

Reviewer #2 (Significance (Required)):This study showed that Dg genetically interact with individual Exocyst complex components, including Sec6, Sec10 and Sec15. Some data presented in the manuscript require further validation. Given the fact that Dg is required to be transported intracellularly to its target membrane component for function, the genetic interactions between Dg and exocyst complex is not novel. If the authors can further illustrate the roles of individual components in Dg interactions/trafficking, the overall impact of this study will be much enhanced.Compare to existing published knowledge:A similar approach was employed by the authors to address muscle-specific dystroglycan interactome. This is a sound approach to identify additional players in Dg-specific pathways in different tissues/cell types.

[Editors' note: further revisions were suggested prior to acceptance, as described below.]

Essential revisions:1) Expand discussion of mass-spec interactome, including limitations of the analyses. As pointed out by one reviewer, the mass-spec results would be strengthened by biochemical evidence for interaction between Dg and the proteins identified by mass-spec (such as co-IP using antibodies specific for Dg interactors). The study nonetheless does present extensive genetic analyses for one group of interactors, components of the exocyst complex, and in my evaluation these genetic interaction experiments provide sufficient functional evidence of a regulatory role for the overall point of this work. What is lacking, however, is discussion of the other groups that were identified. Rather than perform new experiments, the authors should explicitly discuss the elements of protein interaction network presented in 4D, including any known links to Dg or Dg-related disease and limitations/caveats of the mass-spec approach. Such a discussion will provide the reader with valuable context about the Dg interactome.

We thank the editor and the reviewer for this suggestion. Now we have added discussion regarding the limitation of the mass-spec approach to the Discussion section. We also included a discussion of the results of the Dg interactome. Now in the Results section, we talk about the potential significance of Dg interaction with the main elements of the protein interaction network as it relates to Dg functions and human disease development.

2) Clarify some aspects of the exocyst-Dg interaction. The authors present data showing intracellular accumulation of Dg in exocyst mutant cells from salivary glands as well as various cell types in the brain, which demonstrates that exocyst regulation of Dg is not neuron-specific. Unadressed, however, is the key question of whether exocyst affects Dg through its general function in the secretory pathway or through a Dg-specific mechanism. Can the authors provide evidence one way or another, for instance by immunostaining sec mutants for other cell-surface ECM receptors or transmembrane proteins, or (given eLife's COVID response), provide an argument for a general or specific scenario?

We thank the editor and the reviewer for this suggestion. We analyzed the distribution of various cell adhesion proteins: Discs large, Dlg; Armadillo, Arm; DE-Cadherin, Cad; Fasciclin II, FasII; and two integrins, Multiple edematous wings, Mew (PS1) and Inflated, If (PS2). Surprisingly, we found that the localization of these proteins was not altered in clones lacking one of the Sec proteins (new Figure 5—figure supplement 3). The only protein that displayed aberrant localization in *sec15* clones in the brain was Fas2, implying that exocyst-mediated trafficking affects only a subset of proteins in the cell type assayed. One explanation for such specificity could be that both Dg and Fas2 proteins are glycosylated in the process of posttranslation modification (Dempsey et al., 2019; Nakamura et al., 2010; Parkinson et al., 2013; Patel et al., 1987; Snow et al., 1989), while all other proteins tested here have not been shown to be glycosylated. These results are in agreement with a previously published report that studied the specific effect of the exocyst complex on the localization of various transmembrane cell adhesion and cell signaling proteins required for proper photoreceptor development (Mehta et al., 2005). Together, our new data demonstrate that the function of the exocyst complex in Dg trafficking is rather specific and suggest that it might include transport of other proteins undergoing glycosylation.

A second issue is that, in Figure 5C and 5D, it is difficult to distinguish cell-surface and intracellular Dg in the sec-mutant cells in larval brain. These data are central to the author's model that failed delivery of Dg to the surface of these cells leads to neuropil defects, but the small size and minimal cytoplasmic volume makes it hard for me to interpret these data on their own merits. Can the authors provide additional support for the cellular localization of Dg in larval brains-perhaps, for instance, by performing immunostaining of unpermeabilized tissue, or by providing other experimental evidence or (given eLife's COVID response) scientific rationales for the localization of Dg in these cells?

We agree that the images we have provided have not allowed to distinguish cell-surface and intracellular Dg; therefore, we have redone the experiment. The new image in Figure 5E clearly shows that Dg is enriched in cytoplasmic foci and not properly delivered to the membranes of neuroblasts and their progeny. We are thankful for the suggestion of this experiment.